# CR-CTC: CONSISTENCY REGULARIZATION ON CTC FOR IMPROVED SPEECH RECOGNITION

**Zengwei Yao, Wei Kang, Xiaoyu Yang, Fangjun Kuang, Liyong Guo, Han Zhu,**
**Zengrui Jin, Zhaoqing Li, Long Lin, Daniel Povey**
Xiaomi Corp., Beijing, China
`dpovey@xiaomi.com`

## ABSTRACT

Connectionist Temporal Classification (CTC) is a widely used method for automatic speech recognition (ASR), renowned for its simplicity and computational efficiency. However, it often falls short in recognition performance. In this work, we propose the Consistency-Regularized CTC (CR-CTC), which enforces consistency between two CTC distributions obtained from different augmented views of the input speech mel-spectrogram. We provide in-depth insights into its essential behaviors from three perspectives: 1) it conducts self-distillation between random pairs of sub-models that process different augmented views; 2) it learns contextual representation through masked prediction for positions within time-masked regions, especially when we increase the amount of time masking; 3) it suppresses the extremely peaky CTC distributions, thereby reducing overfitting and improving the generalization ability. Extensive experiments on LibriSpeech, Aishell-1, and GigaSpeech datasets demonstrate the effectiveness of our CR-CTC. It significantly improves the CTC performance, achieving state-of-the-art results comparable to those attained by transducer or systems combining CTC and attention-based encoder-decoder (CTC/AED). We release our code at `https://github.com/k2-fsa/icefall`.

## 1 INTRODUCTION

End-to-end approaches (Graves et al., 2006; Graves, 2012; Chan et al., 2015), which eliminate the need of pre-aligned speech-text data, have replaced traditional hybrid systems (Bourlard & Morgan, 2012; Hinton et al., 2012) and become dominant methods in automatic speech recognition (ASR). Prominent examples include Connectionist Temporal Classification (CTC) (Graves et al., 2006), Transducer (Graves, 2012) (also known as RNN-T), and the method that combines CTC and attention-based encoder-decoder (AED) (Chan et al., 2015), referred to as CTC/AED (Watanabe et al., 2017). To handle the alignment between speech and token sequences, CTC (Graves et al., 2006) introduces a blank token and makes independent predictions at each frame, training the model to maximize the total probability over all valid alignments. Transducer (Graves, 2012) extends CTC by introducing a prediction network and a joint network, explicitly modeling the interdependencies on output labels. CTC/AED (Watanabe et al., 2017) integrates CTC into AED (Chan et al., 2015) for jointly training, while the CTC and AED scores are fused during the decoding process. Among these three methods, CTC is the simplest and most computationally efficient due to its frame-independent assumption, making it a strong candidate for real-world deployment. However, it significantly lags behind transducer and CTC/AED in terms of recognition performance, which limits its applicability.

To improve the CTC performance, in this work we propose the Consistency-Regularized CTC (*CR-CTC*), which takes two different augmented views of the same speech mel-spectrogram as input, and enforces consistency between the resulting CTC distributions. We analyze its internal behaviors from three following perspectives. First, it performs self-distillation between sub-models randomly sampled by drop-based techniques (Srivastava et al., 2014; Huang et al., 2016). Second, for positions within time-masked regions, the model is required to predict the target token distributions, forcing it to learn contextual representation based on unmasked context, akin to self-supervised learning methods (Devlin et al., 2019; Baevski et al., 2020; Hsu et al., 2021). Therefore, we especially increase the amount of time masking in *CR-CTC* to enhance this masked prediction behavior. Third, the consistency regularization suppresses extremely peaky CTC distributions, which mitigates over-

fitting and improves the model's generalization ability. Inspired by this, we additionally propose an simple method specifically designed to learn smoother CTC distributions (Appendix Section A.1), which is experimentally validated to be effective.

We conduct experiments on LibriSpeech, Aishell-1, and GigaSpeech datasets using Zipformer (Yao et al., 2024) as speech encoder. The results demonstrate the superiority of *CR-CTC*, which significantly outperforms vanilla CTC and achieves results comparable to, or even slightly better than, those of transducer and CTC/AED. In addition, *CR-CTC* can further improve the performance of transducer and CTC/AED when employed for jointly training. We perform detailed ablation studies on LibriSpeech dataset to investigate the effect of each functional component in *CR-CTC* and to validate our explanations.

## 2 RELATED WORK

**Self-distillation.** Unlike traditional knowledge distillation (Buciluǎ et al., 2006; Hinton et al., 2015), which transfers knowledge from a larger and high-capacity teacher model to a smaller student model, self-distillation (Furlanello et al., 2018; Zhu et al., 2018; Mobahi et al., 2020; Allen-Zhu & Li, 2020) involves learning from a same-architecture model that processes the same training data. This approach enables the model to extract more refined representations and achieve improved performance. For example, BANs (Furlanello et al., 2018) introduces a re-training procedure in which a newly initialized student model is trained to match a pre-trained teacher model, subsequently serving as the teacher in the next iteration. Some works also explore constructing the teacher and student models from a shared network, distilling knowledge from deeper layers to shallower layers (Zhang et al., 2019; Kim et al., 2024), or between pairs of sub-models randomly initialized through drop-based techniques (Srivastava et al., 2014; Huang et al., 2016), such as R-Drop (Wu et al., 2021) and cosub (Touvron et al., 2023). Our *CR-CTC* fundamentally conducts self-distillation between random sub-models, sharing similar idea to R-Drop and cosub, while our approach further use different augmented input views, which enriches the diversity of predictions from these sub-models.

**Masked prediction.** Masked prediction has proven highly effective in self-supervised learning (Devlin et al., 2019; Baevski et al., 2019; Joshi et al., 2020; Baevski et al., 2020; Hsu et al., 2021; He et al., 2022; Baevski et al., 2023). In this approach, the model is tasked with predicting masked positions based on the surrounding unmasked context, which encourages the learning of robust contextual representations. Notable methods for speech representation learning include wav2vec 2.0 (Baevski et al., 2020), HuBERT (Hsu et al., 2021), and data2vec 2.0 (Baevski et al., 2023), which primarily differ in their prediction targets. Specifically, wav2vec 2.0 (Baevski et al., 2020) jointly trains a representation quantizer and learns to distinguish the true quantized target from distractors (Oord et al., 2018). HuBERT generates target labels through offline clustering, while data2vec 2.0 uses contextualized representations from a teacher model as its targets. Our *CR-CTC* essentially performs masked prediction for positions within time-masked regions, where the target labels are frame-level token distributions generated based on another augmented view of input.

**Peaky CTC distributions.** CTC models are known for predicting extremely peaky distributions (Graves et al., 2006; Sak et al., 2015), which can be harmful in certain scenarios, such as forced alignment (Huang et al., 2024) and knowledge distillation (Ding et al., 2020). These peaky distributions lead to inaccurate alignments as the model assigns excessive blanks to non-silence frames. To address this, label priors are employed to suppress the peaky distributions, thereby improving the accuracy of forced alignment (Huang et al., 2024). As position mismatches of CTC spikes can hinder knowledge distillation performance, some approaches propose to encourage consistent alignments between the teacher and student (Ding et al., 2020) or to utilize sequence-level distillation (Takashima et al., 2019). Unlike previous works, we demonstrate that peak suppression in *CR-CTC* can improve the generalization ability of the CTC models.

**Consistency regularization.** The technique of consistency regularization has demonstrated effectiveness in learning generalizable image representations across various learning paradigms, including self-supervised (Chen et al., 2020; Grill et al., 2020; He et al., 2020; Chen & He, 2021), semi-supervised (Sajjadi et al., 2016; Laine & Aila, 2016; Sohn et al., 2020), and supervised (Wu et al., 2021; Touvron et al., 2023; Heo et al., 2023) learning tasks. Self-supervised methods, such as Sim-CLR (Chen et al., 2020), BYOL (Grill et al., 2020), MoCo (He et al., 2020) and SimSiam (Chen & He, 2021), aim to align hidden representations of unlabeled image data from different model branches or different augmented views. They address the training issue of feature collapsing into a

constant vector (Chen & He, 2021) through contrastive learning (Chen et al., 2020; He et al., 2020), momentum encoder (Grill et al., 2020; He et al., 2020), and stop-gradient operation (Chen & He, 2021). In semi-supervised learning, a prominent example leveraging consistency regularization is FixMatch (Sohn et al., 2020). It generates pseudo-labels based on high-confidence predictions from weakly augmented images, then trains the model to predict these pseudo-labels using the strongly augmented versions of the same images. Additionally, in supervised learning, methods such as R-Drop (Wu et al., 2021) and cosub (Touvron et al., 2023) encourage consistency between predictions of randomly sampled sub-models using drop-based techniques.

When employing consistency regularization as unsupervised objective to train transformer encoders on unlabeled speech data, a new training issue arises in the form of the shortcut learning problem (Geirhos et al., 2020), which is tackled using reconstruction loss in Speech SimCLR (Jiang et al., 2020) and temporal augmentation in C-Siam (Khorram et al., 2022). Some studies explore leveraging consistency regularization to enhance model robustness during predicting the pseudo-labels of untranscribed data, which are generated based on different augmentations (Masumura et al., 2020; Weninger et al., 2020; Chen et al., 2021b; Higuchi et al., 2021; Sapru, 2022) or through speech chain reconstruction (Qi et al., 2022). In contrast to these self/semi-supervised ASR works, our work focuses on a fully supervised setting, where we introduce consistency loss as a regularization term to improve performance of CTC model trained on labeled data. As the consistency regularization is enforced on CTC distributions, which are stably supervised by the main CTC loss, it inherently avoids the training issues associated with the unsupervised objectives as observed in Speech Sim-CLR (Jiang et al., 2020) and C-Siam (Khorram et al., 2022).

The idea of R-Drop (Wu et al., 2021) has also been extended to supervised ASR (Gao et al., 2022; Yoon et al., 2024). For example, to improve the CTC/AED system, (Gao et al., 2022) specially designs the spatial-temporal dropout to construct the sub-models, with consistency regularization enforced exclusively on the CTC spike frames. Cons-KD (Yoon et al., 2024) integrates consistency regularization into a knowledge distillation system, enabling the student model to be more robust to inconsistency induced by dropout. In this work, we focus on improving the performance of pure CTC systems and are the first to enable CTC models to match the performance of transducer and CTC/AED systems by a simple yet effective approach. Moreover, we introduce peak suppression as a novel explanatory perspective, demonstrating for the first time that it can mitigate overfitting and enhance the generalization ability of CTC models.

## 3 METHOD

We first introduce the standard CTC algorithm in Section 3.1. Then we present the detailed implementation of our proposed Consistency-Regularized CTC (*CR-CTC*) in Section 3.2, followed by in-depth explanations from different perspectives in Section 3.3.

### 3.1 PRELIMINARY: CONNECTIONIST TEMPORAL CLASSIFICATION

The ASR task is to convert a sequence of speech frames $\mathbf{x} = \{x_t\}_1^T$ of length $T$ to a sequence of transcript tokens $\mathbf{y} = \{y_u \in \mathcal{V}\}_1^U$ of length $U$, where $\mathcal{V}$ is the vocabulary and typically $T \geq U$. CTC (Graves et al., 2006) extends the vocabulary $\mathcal{V}$ to $\mathcal{V}' = \mathcal{V} \cup \{\epsilon\}$ with a blank token $\epsilon$, and aims to maximize the total posterior probability of all valid alignments $\boldsymbol{\pi} = \{\pi_t \in \mathcal{V}'\}_1^T$ between $\mathbf{x}$ and $\mathbf{y}$. Let $\mathcal{B}(\boldsymbol{\pi})$ denote the many-to-one map that merges repeating tokens and removes all blanks in $\boldsymbol{\pi}$, and $p(\boldsymbol{\pi}|\mathbf{x})$ denote the posterior probability of alignment $\boldsymbol{\pi}$, the CTC loss function is formulated as:

$$\mathcal{L}_{\mathrm{CTC}}(\mathbf{x}, \mathbf{y}) = -\log \sum_{\boldsymbol{\pi} \in \mathcal{B}^{-1}(\mathbf{y})} p(\boldsymbol{\pi}|\mathbf{x}). \tag{1}$$

Specifically, given the input $\mathbf{x}$, it employs an encoder $f$ to estimate the $|\mathcal{V}'|$-dimensional probability distributions $\mathbf{z} = \{z_t\}_1^T$: $\mathbf{z} = f(\mathbf{x})$ [1], where $f$ is modeled by a speech encoder network such as Zipformer (Yao et al., 2024) followed by a linear projection layer and a *softmax* function. Note that we now start to use $\mathcal{L}_{\mathrm{CTC}}(\mathbf{z}, \mathbf{y})$ instead of $\mathcal{L}_{\mathrm{CTC}}(\mathbf{x}, \mathbf{y})$ for ease of description in the following sections. Under the frame-independent assumption (Graves et al., 2006), $p(\boldsymbol{\pi}|\mathbf{x})$ is computed as:

$$p(\boldsymbol{\pi}|\mathbf{x}) = \prod_{t=1}^{T} z_{t,\pi_t}, \tag{2}$$

---

[1]$T$ is typically downsampled in the encoder $f$ by a factor of 4 for efficiency. This is omitted for the sake of simplicity in expression.

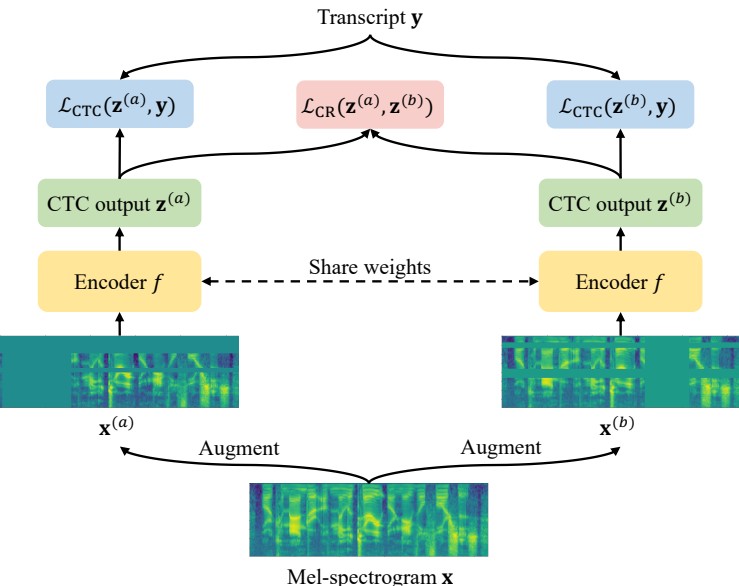

Figure 1: Overall architecture of *CR-CTC*.

where $z_{t,\pi_t}$ is the probability of emitting token $\pi_t$ at frame $t$.

## 3.2  OUR APPROACH: CONSISTENCY-REGULARIZED CTC

Figure 1 illustrates the overall architecture of our proposed *CR-CTC*. It takes as input two different augmented views, $\mathbf{x}^{(a)}$ and $\mathbf{x}^{(b)}$, both derived from the input speech mel-spectrogram $\mathbf{x}$. The two input views are then passed through a shared speech encoder $f$, which estimates the per-frame distributions: $\mathbf{z}^{(a)} = f(\mathbf{x}^{(a)})$, $\mathbf{z}^{(b)} = f(\mathbf{x}^{(b)})$. In addition to computing the CTC losses on both branches: $\mathcal{L}_{\text{CTC}}(\mathbf{z}^{(a)}, \mathbf{y})$ and $\mathcal{L}_{\text{CTC}}(\mathbf{z}^{(b)}, \mathbf{y})$, we introduce an auxiliary loss (defined in Equation 4) to enforce consistency between $\mathbf{z}^{(a)}$ and $\mathbf{z}^{(b)}$: $\mathcal{L}_{\text{CR}}(\mathbf{z}^{(a)}, \mathbf{z}^{(b)})$. The overall loss of the whole model is formulated as:

$$\mathcal{L} = \frac{1}{2}(\mathcal{L}_{\text{CTC}}(\mathbf{z}^{(a)}, \mathbf{y}) + \mathcal{L}_{\text{CTC}}(\mathbf{z}^{(b)}, \mathbf{y})) + \alpha \mathcal{L}_{\text{CR}}(\mathbf{z}^{(a)}, \mathbf{z}^{(b)}), \tag{3}$$

where $\alpha$ is a hyper-parameter that controls the consistency regularization.

**Different augmented views.** The two different augmented views, $\mathbf{x}^{(a)}$ and $\mathbf{x}^{(b)}$, are generated by independently applying SpecAugment (Park et al., 2019) to two copies of the input mel-spectrogram $\mathbf{x}$. SpecAugment involves warping along time axis, masking blocks of frequency channels, and masking blocks of time steps. Since time warping alters feature timing and thus shifts output timestamps, we apply it first before creating the copies to prevent significant timestamp mismatches between the outputs of two branches. Subsequently, random frequency masking and time masking are both applied to the two copies, resulting in $\mathbf{x}^{(a)}$ and $\mathbf{x}^{(b)}$. Note that we also increase the amount of time masking by a factor of 2.5 compared to regular systems. The reason behind this adjustment is explained in Section 3.3, with implementation details provided in Section 4.1.

**Consistency regularization loss.** The consistency regularization is applied on each frame $t$, by minimizing the bidirectional Kullback-Leibler divergence (denoted as $D_{\text{KL}}$) between each pair of distributions $z_t^{(a)}$ and $z_t^{(b)}$: $D_{\text{KL}}(sg(z_t^{(b)})\|z_t^{(a)})$ and $D_{\text{KL}}(sg(z_t^{(a)})\|z_t^{(b)})$, where $sg$ denotes the operation stopping gradient on the target distributions. The regularization loss $\mathcal{L}_{\text{CR}}(\mathbf{z}^{(a)}, \mathbf{z}^{(b)})$ is formulated as:

$$\mathcal{L}_{\text{CR}}(\mathbf{z}^{(a)}, \mathbf{z}^{(b)}) = \frac{1}{2}\sum_{t=1}^{T} D_{\text{KL}}(sg(z_t^{(b)})\|z_t^{(a)}) + D_{\text{KL}}(sg(z_t^{(a)})\|z_t^{(b)}). \tag{4}$$

### 3.3 EXPLANATION

We now explain the essential behaviors of our proposed *CR-CTC* from three different perspectives: 1) it performs self-distillation between pairs of sub-models with different input views; 2) it conducts contextual representation learning by predicting the token distributions at masked positions based on unmasked context; 3) it suppresses extremely peaky CTC distributions, mitigating overfitting and enhancing generalization ability. We conduct an empirical investigation through ablation studies in Section 4.3, and the experimental results validate our explanations.

**Self-distillation.** When using model regularization techniques such as dropout (Srivastava et al., 2014) and stochastic depth (Huang et al., 2016), which randomly drop parts of the model (neurons or layers), it can be viewed as implicitly training randomly sampled sub-models that are ultimately combined into an ensemble during inference. Similar to R-Drop (Wu et al., 2021) and cosub (Touvron et al., 2023), in *CR-CTC*, enforcing consistency regularization between the two branches enables to perform self-distillation between pairs of randomly sampled sub-models derived from the shared model $f$, with each sub-model receiving supervision signals in the form of per-frame predictions from the other. In addition, feeding different augmented views (with larger amount of time masking) exposes these sub-models to varied aspects of the input data, enhancing their prediction diversity and facilitating richer knowledge transfer as well as complementary representation learning.

**Masked prediction.** In *CR-CTC*, consistency regularization requires frames within the time-masked regions in each branch to predict the corresponding token distributions, which are generated by the other branch on the fly. Similar to masked-based self-supervised models (Devlin et al., 2019; Baevski et al., 2020; Hsu et al., 2021), this behavior encourages the model to capture acoustic information on the unmasked context and exploit its implicit language modeling capability. Independently applying random time masking to the two branches reduces the occurrence of positions masked by both branches, thereby improve the quality of the provided target distributions for these masked positions. Furthermore, increasing the amount of time masking in *CR-CTC* enhances contextual representation learning through the masked prediction behavior.

**Peak suppression.** In line with previous works (Graves et al., 2006; Sak et al., 2015), we also observe that CTC tends to learn extremely peaky distributions. As shown in Figure 2 (left), almost all non-blank tokens occupy only one frame, while the remaining frames are dominated by the blank token, with both types of emissions occurring with extremely high probabilities. This phenomenon suggests potential overfitting to training data, which limits generalization ability to unseen data.

Enforcing prediction consistency between the two branches in *CR-CTC* guides the model to learn the average of their predictions, ultimately resulting in smoother distributions. The peak suppression behavior reduces overconfidence on training data, thereby improving the model's generalization ability. As presented in Figure 2 (right), *CR-CTC* exhibits reduced token emitting probabilities and an increased occurrence of repeating non-blank tokens. A comparison of concrete statistics on the distribution peakedness between CTC and *CR-CTC* is provided in Table 6.

Inspired by this, we also propose a simple method, called Smooth-Regularized CTC (*SR-CTC*), which incorporates an auxiliary loss into regular CTC, specifically encouraging the model to learn smoother CTC distributions. Appendix Section A.1 presents the details of *SR-CTC*.

## 4 EXPERIMENTS

### 4.1 EXPERIMENTAL SETUP

**Datasets.** To evaluate the effectiveness of our proposed *CR-CTC*, we conduct experiments on three publicly available ASR datasets: 1) LibriSpeech (Panayotov et al., 2015), which contains 1000 hours of English speech; 2) Aishell-1 (Bu et al., 2017), which consists of 170 hours of Mandarin speech; 3) GigaSpeech (Chen et al., 2021a), comprising 10000 hours of English speech.

**Implementation details.** Our experiments are performed using the icefall framework [2], with Lhotse toolkit (Żelasko et al., 2021) for data preparation. For regular ASR recipes in icefall, default parameter settings of SpecAugment (Park et al., 2019) include a time warping factor of 80, 2 frequency

---

[2] `https://github.com/k2-fsa/icefall`

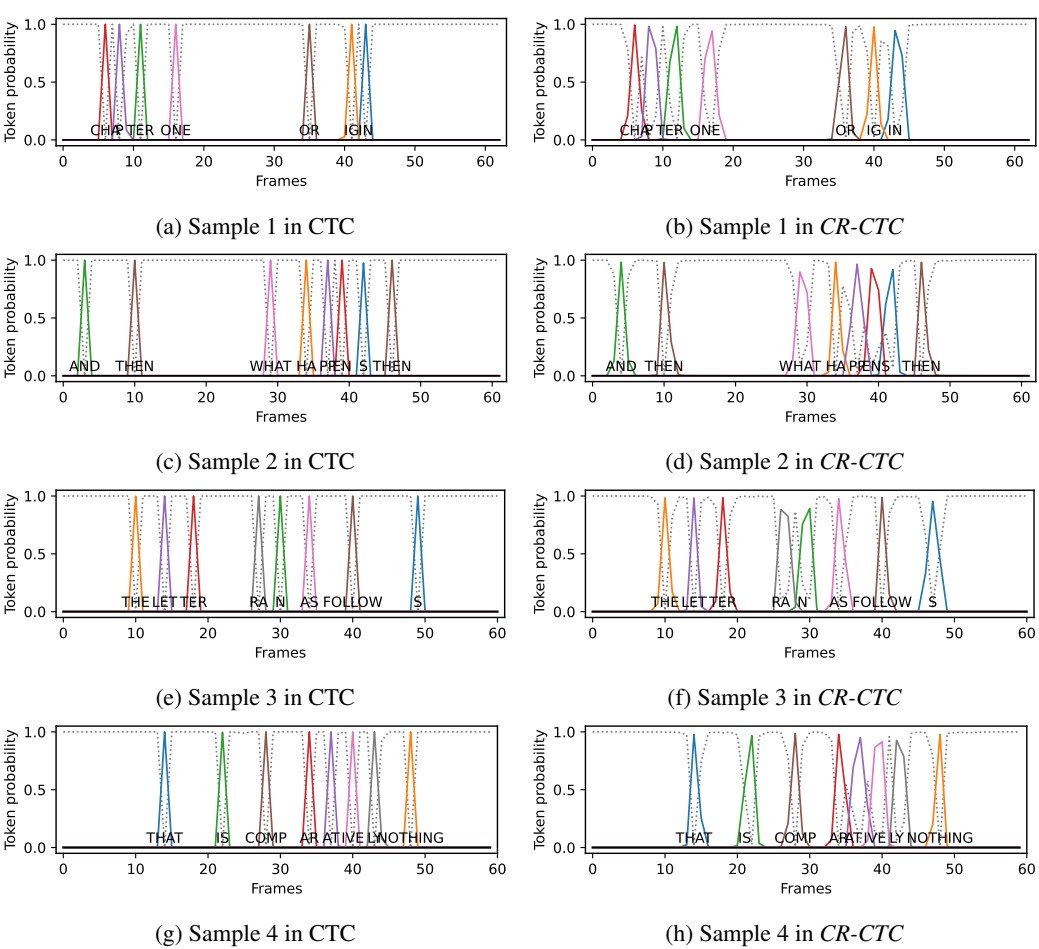

Figure 2: Visualization of token emitting probabilities for vanilla CTC (left) and our *CR-CTC* (right) on four randomly selected samples from LibriSpeech test set. The gray dashed lines indicate the blank token. Compared to vanilla CTC, the token distributions in *CR-CTC* are smoother with lower emitting probabilities and more repeating non-blank tokens.

masking regions with a maximum width of 27, and 10 time masking regions with a maximum width of 100, along with a maximum masking fraction of 15% specifically for time masking [3]. In our *CR-CTC* systems, we utilize larger amount of time masking through increasing both the number of time masking regions and the maximum masking fraction by a factor of 2.5. Speed perturbation (Ko et al., 2015) with factors 0.9, 1.0 and 1.1 is applied to LibriSpeech (Panayotov et al., 2015) and Aishell-1 (Bu et al., 2017) datasets. The input features are 80-dimensional mel-spectrograms extracted using 25-ms windows with a 10-ms shift. For LibriSpeech and GigaSpeech datasets, we employ 500-class Byte Pair Encoding (BPE) (Sennrich et al., 2016) word pieces as modeling units, while for Aishell-1 dataset, we use 4336-class characters. By default, we set $\alpha$ in Equation 3 to 0.2. Zipformer (Yao et al., 2024), which uses dropout (Srivastava et al., 2014) and stochastic depth (Huang et al., 2016), is used as our speech encoder due to its speed and high performance. It takes input features at frame rate of 100Hz, processes the sequence through 6 stacks with frame rates of 50Hz, 25Hz, 12.5Hz, 6.25Hz, 12.5Hz, and 25Hz, and finally produces the encoder output at frame rate of 25Hz. Following (Yao et al., 2024), pruned transducer (Kuang et al., 2022), a highly optimized and memory-efficient version of transducer, is employed for comparison. Word-error-rate (WER) and character-error-rate (CER) are employed as ASR metrics for English and Mandarin datasets, respectively. As *CR-CTC* requires two forward pass during training, we train *CR-CTC* models with half the batch size and half the number of epochs compared to CTC models, ensuring a fair com-

---

[3]See the SpecAugment implementation in Lhotse for more details: `https://github.com/lhotse-speech/lhotse/blob/master/lhotse/dataset/signal_transforms.py`

parison in terms of training cost. Training configuration in terms of number of GPUs and training epochs are provided in Appendix Section A.2. For CTC and *CR-CTC* systems, we use prefix search decoding (Graves et al., 2006) with a beam size of 4 for comparisons against other state-of-the-art models, and employ greedy search decoding for ablation studies. Results comparison between these two decoding methods are provided in Appendix Section A.3. For pruned transducer models, we use beam search decoding with beam size of 4 (Kang et al., 2023). For CTC/AED systems, we use joint decoding that combines CTC scores and AED scores (Watanabe et al., 2017).

## 4.2 COMPARISON WITH STATE-OF-THE-ART MODELS

In this section, we compare our *CR-CTC* with other state-of-the-art models. For LibriSpeech and GigaSpeech datasets, we also use *CR-CTC* as an auxiliary loss in CTC/AED and pruned transducer systems for joint training (denoted as *CR-CTC*/AED and pruned transducer w/ *CR-CTC*), to further validate the representation learning capability of *CR-CTC*. Note that for the models that combine *CR-CTC* and pruned transducer, we only utilize the transducer head for decoding, without incorporating the CTC scores. For the larger GigaSpeech dataset, we additionally use a even larger scale of Zipformer (Zipformer-XL). Model configuration of different scales of Zipformer are provided in Table 15. For Aishell-1 dataset, which is considerably smaller, we conduct experiments on Zipformer-S and Zipformer-M to ensure comparable parameter counts with other models reported in the literature.

Table 1: WER(%) performance of our method on LibriSpeech dataset compared to the best results reported in the literature without using an external language model.

| Model | Params (M) | WER (%) | |
| --- | --- | --- | --- |
| | | *test-clean* | *test-other* |
| CTC/AED, E-Branchformer-B (Kim et al., 2023) | 41.1 | 2.49 | 5.61 |
| CTC/AED, Branchformer (Peng et al., 2022) | 116.2 | 2.4 | 5.5 |
| CTC/AED, E-Branchformer-L (Kim et al., 2023) | 148.9 | 2.14 | 4.55 |
| Transducer, ContextNet-S (Han et al., 2020) | 10.8 | 2.9 | 7.0 |
| Transducer, ContextNet-M (Han et al., 2020) | 31.4 | 2.4 | 5.4 |
| Transducer, ContextNet-L (Han et al., 2020) | 112.7 | 2.1 | 4.6 |
| Transducer, Conformer-S (Gulati et al., 2020) | 10.3 | 2.7 | 6.3 |
| Transducer, Conformer-M (Gulati et al., 2020) | 30.7 | 2.3 | 5.0 |
| Transducer, Conformer-L (Gulati et al., 2020) | 118.8 | 2.1 | 4.3 |
| Transducer, MH-SSM 32L (Fathullah et al., 2023) | 140.3 | 2.01 | 4.61 |
| Transducer, Stateformer 25L (Fathullah et al., 2023) | 139.8 | 1.91 | 4.36 |
| CTC/AED, Zipformer-S (Yao et al., 2024) | 46.3 | 2.46 | 6.04 |
| CTC/AED, Zipformer-M (Yao et al., 2024) | 90.0 | 2.22 | 4.97 |
| CTC/AED, Zipformer-L (Yao et al., 2024) | 174.3 | 2.09 | 4.59 |
| Pruned transducer, Zipformer-S (Yao et al., 2024) | 23.3 | 2.42 | 5.73 |
| Pruned transducer, Zipformer-M (Yao et al., 2024) | 65.6 | 2.21 | 4.79 |
| Pruned transducer, Zipformer-L (Yao et al., 2024) | 148.4 | 2.00 | 4.38 |
| CTC, Zipformer-S | 22.1 | 2.85 | 6.89 |
| CTC, Zipformer-M | 64.3 | 2.52 | 6.02 |
| CTC, Zipformer-L | 147.0 | 2.5 | 5.72 |
| *CR-CTC*, Zipformer-S (ours) | 22.1 | 2.52 | 5.85 |
| *CR-CTC*, Zipformer-M (ours) | 64.3 | 2.1 | 4.61 |
| *CR-CTC*, Zipformer-L (ours) | 147.0 | 2.02 | 4.35 |
| *CR-CTC*/AED, Zipformer-L (ours) | 174.3 | 1.96 | 4.08 |
| Pruned transducer w/ *CR-CTC*, Zipformer-L (ours) | 148.8 | **1.88** | **3.95** |

**LibriSpeech dataset.** Table 1 presents the results on LibriSpeech dataset for *CR-CTC* and other state-of-the-art models. Our *CR-CTC* significantly outperforms the CTC baselines on all three scales of Zipformer encoder. When comparing to CTC/AED models, our *CR-CTC* achieves lower WER on Zipformer-M/L, while yielding comparable result on Zipformer-S. Similarly, our *CR-CTC* surpasses pruned transducer on Zipformer-M, and performs comparably on Zipformer-L. It also demonstrates that *CR-CTC* can further enhance the performance of CTC/AED and pruned transducer models when used for jointly training. A notable result is that pruned transducer combined with *CR-CTC*

using Zipformer-L achieves a new state-of-the-art result of 1.88%/3.95% on *test-clean/test-other*, outperforming both the transducer models with Conformer-L (Gulati et al., 2020) and Stateformer 25L (Fathullah et al., 2023).

Table 2: WER(%) performance of our method on Aishell-1 dataset compared to the best results reported in the literature without using an external language model.

| Model | Params (M) | WER (%) dev | test |
|---|---|---|---|
| CTC/AED, Conformer in ESPnet (Watanabe et al., 2018) | 46.2 | 4.5 | 4.9 |
| CTC/AED, Conformer in WeNet (Yao et al., 2021) | 46.3 | – | 4.61 |
| CTC/AED, E-Branchformer in ESPnet (Watanabe et al., 2018) | 37.9 | 4.2 | 4.5 |
| CTC/AED, Branchformer (Peng et al., 2022) | 45.4 | 4.19 | 4.43 |
| Pruned transducer, Zipformer-S (Yao et al., 2024) | 30.2 | 4.4 | 4.67 |
| Pruned transducer, Zipformer-M (Yao et al., 2024) | 73.4 | 4.13 | 4.4 |
| CTC, Zipformer-S | 23.1 | 4.89 | 5.26 |
| CTC, Zipformer-M | 66.2 | 4.47 | 4.8 |
| CTC/AED, Zipformer-S | 39.3 | 4.47 | 4.8 |
| CTC/AED, Zipformer-M | 83.2 | 4.0 | 4.32 |
| *CR-CTC*, Zipformer-S (ours) | 23.1 | 3.9 | 4.12 |
| *CR-CTC*, Zipformer-M (ours) | 66.2 | **3.72** | **4.02** |

**Aishell-1 dataset.** Table 2 presents the results on Aishell-1 dataset. Our *CR-CTC* models not only significantly outperform vanilla CTC by a substantial margin but also achieve better results than all other CTC/AED and pruned transducer models. For example, *CR-CTC* with Zipformer-S surpasses CTC/AED with Zipformer-M while using much fewer parameters.

Table 3: WER(%) performance of our method on GigaSpeech dataset compared to the best results reported in the literature without using an external language model.

| Model | Params (M) | WER (%) dev | test |
|---|---|---|---|
| CTC/AED, Transformer (Chen et al., 2021a) | 87 | 12.30 | 12.30 |
| CTC/AED, Conformer in Wenet (Zhang et al., 2022) | 113.2 | 10.7 | 10.6 |
| CTC/AED, Conformer in ESPnet (Chen et al., 2021a) | 113.2 | 10.9 | 10.8 |
| CTC/AED, E-Branchformer in ESPnet (Watanabe et al., 2018) | 148.9 | 10.6 | 10.5 |
| CTC, Zipformer-S | 22.1 | 12.08 | 11.95 |
| CTC, Zipformer-M | 64.3 | 11.23 | 11.27 |
| CTC, Zipformer-L | 147.0 | 11.16 | 11.16 |
| CTC, Zipformer-XL | 286.6 | 10.8 | 10.87 |
| CTC/AED, Zipformer-S | 46.3 | 11.4 | 11.39 |
| CTC/AED, Zipformer-M | 90.0 | 10.57 | 10.61 |
| CTC/AED, Zipformer-L | 174.3 | 10.26 | 10.38 |
| CTC/AED, Zipformer-XL | 315.5 | 10.22 | 10.33 |
| Pruned transducer, Zipformer-S | 23.3 | 10.98 | 10.94 |
| Pruned transducer, Zipformer-M | 65.6 | 10.37 | 10.42 |
| Pruned transducer, Zipformer-L | 148.4 | 10.23 | 10.28 |
| Pruned transducer, Zipformer-XL | 288.2 | 10.09 | 10.2 |
| *CR-CTC*, Zipformer-S (ours) | 22.1 | 11.68 | 11.58 |
| *CR-CTC*, Zipformer-M (ours) | 64.3 | 10.62 | 10.72 |
| *CR-CTC*, Zipformer-L (ours) | 147.0 | 10.31 | 10.41 |
| *CR-CTC*, Zipformer-XL (ours) | 286.6 | 10.15 | 10.28 |
| *CR-CTC*/AED, Zipformer-XL (ours) | 315.5 | **9.92** | 10.07 |
| Pruned transducer w/ *CR-CTC*, Zipformer-XL (ours) | 286.6 | 9.95 | **10.03** |

**GigaSpeech dataset.** Table 3 shows the results on GigaSpeech dataset. Our *CR-CTC* consistently achieves a significantly lower WER than vanilla CTC across all scales of Zipformer. In comparisons with CTC/AED or pruned transducer models, our *CR-CTC* demonstrates comparable performance

on Zipformer L/XL. Additionally, the results indicate that employing *CR-CTC* for joint training can further improve the performance of both CTC/AED and pruned transducer models.

### 4.3 ABLATION STUDIES

We now perform ablation studies on LibriSpeech dataset using Zipformer-M encoder to investigate the effect of each component in *CR-CTC* (Section 3.2), and to validate our explanations of its behaviors (Section 3.3). Results of tuning $\alpha$ in Equation 3 and the ratio used to increase the amount of time masking are presented in Table 16.

Table 4: Ablation studies for self-distillation in *CR-CTC* on LibriSpeech dataset using Zipformer-M encoder and greedy search decoding.

| Method | WER (%) | |
| --- | --- | --- |
| | *test-clean* | *test-other* |
| CTC baseline | 2.51 | 6.02 |
| EMA-distilled CTC | 2.31 | 5.25 |
| ***CR-CTC* (final)** | **2.12** | **4.62** |
| No larger time masking | 2.19 | 4.98 |
| No larger time masking, no different augmented views | 2.27 | 5.11 |
| Use hard-label CE-based $\mathcal{L}_{\mathrm{CR}}$ | 2.14 | 4.84 |
| Remove *sg* in $\mathcal{L}_{\mathrm{CR}}$ | 2.24 | 4.97 |

**Self-distillation.** One self-distillation method in self-supervised learning is to construct a teacher model by tracking the model weights using exponential moving average (EMA) (Grill et al., 2020; He et al., 2020; Baevski et al., 2023). For comparison, we include this approach, referred to as EMA-distilled CTC, which incorporates an auxiliary loss to learn from the CTC distribution of the EMA teacher model. Its details are provided in Appendix Section A.6. As presented in Table 4, *CR-CTC* significantly outperforms EMA-distilled CTC, demonstrating its superiority in self-distillation. For *CR-CTC*, both the lack of increased time masking and the absence of different augmented views lead to WER degradation, indicating the effectiveness of enhancing the input diversity between sub-models during self-distillation. Replacing $D_{\mathrm{KL}}$ with hard label-based cross-entropy (CE) function in $\mathcal{L}_{\mathrm{CR}}$ (Equation 4) results in a WER degradation of 0.02%/0.22% on *test-clean/test-other*. This suggests the advantage of using $D_{\mathrm{KL}}$ which enables a finer-grained self-distillation as it distills over the full CTC lattice, whereas the hard label CE-based method only distills the best alignment. When removing the *sg* operation in $\mathcal{L}_{\mathrm{CR}}$, the WER increase by 0.12%/0.35%, which implies that the model might have a tendency towards a degenerated solution (Chen & He, 2021) that is insensitive to the pattern of input masking and model dropout.

Table 5: Ablation studies for masked prediction in *CR-CTC* on LibriSpeech dataset using Zipformer-M encoder and greedy search decoding.

| Method | WER (%) | |
| --- | --- | --- |
| | *test-clean* | *test-other* |
| CTC baseline | 2.51 | 6.02 |
| Use larger time masking | 2.68 | 6.28 |
| ***CR-CTC* (final)** | **2.12** | **4.62** |
| No larger time masking | 2.19 | 4.98 |
| No larger time masking, no different augmented views | 2.27 | 5.11 |
| No larger time masking, use larger frequency masking | 2.26 | 4.98 |
| Exclude self-masked frames in $\mathcal{L}_{\mathrm{CR}}$ | 2.32 | 5.26 |
| Exclude self-unmasked frames in $\mathcal{L}_{\mathrm{CR}}$ | 2.32 | 5.02 |

**Masked prediction.** As reported in Table 5, without increasing the amount of time masking, the WER of *CR-CTC* increases by 0.07%/0.36% on *test-clean/test-other*, suggesting the effectiveness of enhancing the masked prediction behavior for contextual representation learning. Additionally, without using different augmented views, the WER increases further by 0.12%/0.13%. This indicates the advantage of independently applying random time masking, which improves the quality

of the provided target distributions for the masked positions. However, using larger amount of frequency masking leads to a WER degradation of 0.07% on *test-clean*, implying that the performance gain from increasing the amount of time masking is primarily due to the masked prediction behavior, rather than merely increasing the input diversity for the two branches. Furthermore, applying larger amount of time masking does not benefit the CTC baseline, as it increases the WER by 0.17%/0.26%. In the final *CR-CTC* system, excluding frames with time-masked regions in the current branch (self-masked) from $\mathcal{L}_{CR}$ (Equation 4) leads to a larger WER degradation compared to excluding the remaining unmasked frames (self-unmasked). This highlights the importance of the masked prediction behavior in the overall performance of *CR-CTC*.

Table 6: Ablation studies for peak suppression in *CR-CTC* on LibriSpeech dataset using Zipformer-M encoder and greedy search decoding. We include the averaged duration of all non-blank tokens, as well as the averaged emitting probabilities of the blank token and all non-blank tokens on the best alignments.

| Method | Non-blank duration (frames) | Emit probability (%) | | WER (%) | |
|---|---|---|---|---|---|
| | | blank | non-blank | *test-clean* | *test-other* |
| CTC baseline | 1.04 | 99.64 | 98.50 | 2.51 | 6.02 |
| *SR-CTC* | 4.25 | 95.44 | 90.04 | 2.32 | 5.22 |
| ***CR-CTC*** | 1.28 | 94.19 | 89.42 | **2.12** | **4.62** |

**Peak suppression.** To measure the peakedness of the learned CTC distributions, we compute the averaged duration over all non-blank tokens, as well as the averaged emitting probabilities for the blank token and all non-blank tokens, based on the best alignment obtained through greedy search decoding on the test sets. We also include the method *SR-CTC* (described in Appendix Section A.1) for comparison. As presented in Table 6, compared to the CTC baseline, *CR-CTC* learns smoother distributions and significantly improves the recognition performance. Note that *SR-CTC* also surpasses the CTC baseline by 0.19%/0.8% on *test-clean*/*test-other*, while exhibiting a notably larger average duration of non-blank tokens. This manifests the effectiveness of peak suppression in reducing overfitting and improving generalization performance.

Table 7: Comparison between *CR-CTC* and methods using an auxiliary head for jointly training on LibriSpeech dataset using Zipformer-M encoder and greedy search decoding.

| Method | Params (M) | WER (%) | |
|---|---|---|---|
| | | *test-clean* | *test-other* |
| CTC baseline | 64.3 | 2.51 | 6.02 |
| CTC w/ AED head | 90.0 | 2.46 | 5.57 |
| CTC w/ pruned transducer head | 65.8 | 2.42 | 5.4 |
| ***CR-CTC*** | 64.3 | **2.12** | **4.62** |

**Compared to using auxiliary head for jointly training.** The straightforward approach to improve the CTC performance is using an auxiliary head of AED (Chan et al., 2015; Hentschel et al., 2024) or pruned transducer (Kuang et al., 2022) for jointly training, while retaining only the CTC head for inference. As reported in Table 7, *CR-CTC* significantly outperforms these two methods with less model parameters, suggesting the advantage of our method.

## 5 CONCLUSION

In this work, we introduce the *CR-CTC* to enhance CTC performance. Specifically, it takes as input two different augmented views of the same speech mel-spectrogram, and enforce consistency between the two obtained CTC distributions. We explain our method from three different perspectives: 1) self-distillation between randomly sampled sub-models; 2) masked prediction for positions within time-masked regions, facilitating the learning of contextual representation; 3) peak suppression, which reduces overfitting and improves the model's generalization ability. Extensive experiments on LibriSpeech, Aishell-1, and GigaSpeech datasets demonstrate the effectiveness of *CR-CTC*. Additionally, detailed ablation studies validate our explanations.

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

# A APPENDIX

## A.1 SMOOTH-REGULARIZED CTC

Smooth-regularized CTC (*SR-CTC*) discourages peaky distributions by adding an smooth regularization loss (denoted as $\mathcal{L}_{\mathrm{SR}}$) to regular CTC model. Specifically, we first apply a smooth kernel $K$ of size 3 to the model prediction $\mathbf{z}$, smoothing it along the time dimension: $\mathbf{z}^{(s)} = smooth(\mathbf{z}, K)$. The smoothing operation is done by using a 1-D depth-wise convolution layer. Then we minimize the $D_{\mathrm{KL}}$ between $\mathbf{z}$ and $\mathbf{z}^{(s)}$, similar to the consistency loss in *CR-CTC* (Equation 4):

$$\mathcal{L}_{\mathrm{SR}}(\mathbf{z}, \mathbf{z}^{(s)}) = \sum_{t=1}^{T} D_{\mathrm{KL}}(sg(z_t^{(s)}) \| z_t). \tag{5}$$

The overall loss of *SR-CTC* is formulated as:

$$\mathcal{L}' = \mathcal{L}_{\mathrm{CTC}}(\mathbf{z}, \mathbf{y}) + \beta \mathcal{L}_{\mathrm{SR}}(\mathbf{z}, \mathbf{z}^{(s)}), \tag{6}$$

where $\beta$ is a hyper-parameter. In this work, we use $K = (0.25, 0.5, 0.25)$ and $\beta = 0.2$.

We validate its effectiveness in Section 4.3. Table 6 presents the experimental result.

## A.2 TRAINING CONFIGURATION

Training configuration, including the number of GPUs and training epochs, on LibriSpeech, Aishell-1 and GigaSpeech datasets are presented in Table 8, Table 9, and Table 10, respectively.

Table 8: Training configuration on LibriSpeech dataset.

| Model | GPUs (80G NVIDIA Tesla A100) | Epochs |
|---|---|---|
| CTC, Zipformer-S | 1 | 100 |
| CTC, Zipformer-M | 2 | 100 |
| CTC, Zipformer-L | 2 | 100 |
| *CR-CTC*, Zipformer-S | 1 | 50 |
| *CR-CTC*, Zipformer-M | 2 | 50 |
| *CR-CTC*, Zipformer-L | 2 | 50 |
| *CR-CTC*/AED, Zipformer-L | 2 | 50 |
| Pruned transducer w/ *CR-CTC*, Zipformer-L | 2 | 50 |

Table 9: Training configuration on Aishell-1 dataset.

| Model | GPUs (80G NVIDIA Tesla A100) | Epochs |
|---|---|---|
| CTC, Zipformer-S | 1 | 120 |
| CTC, Zipformer-M | 1 | 120 |
| CTC/AED, Zipformer-S | 1 | 60 |
| CTC/AED, Zipformer-M | 1 | 60 |
| *CR-CTC*, Zipformer-S | 1 | 60 |
| *CR-CTC*, Zipformer-M | 1 | 60 |

## A.3 RESULTS OF DIFFERENT DECODING METHODS

Results comparison between greedy search decoding and prefix search decoding for CTC and *CR-CTC* on LibriSpeech, Aishell-1 and GigaSpeech datasets are presented in Table 11, Table 12, and Table 13, respectively. In addition, Table 14 presents the results of different beam sizes for prefix search decoding on LibriSpeech dataset with Zipformer-M encoder.

Table 10: Training configuration on GigaSpeech dataset.

| Model | GPUs (80G NVIDIA Tesla A100) | Epochs |
|---|---|---|
| CTC, Zipformer-S | 2 | 60 |
| CTC, Zipformer-M | 2 | 60 |
| CTC, Zipformer-L | 2 | 60 |
| CTC, Zipformer-XL | 4 | 60 |
| CTC/AED, Zipformer-S | 2 | 30 |
| CTC/AED, Zipformer-M | 2 | 30 |
| CTC/AED, Zipformer-L | 2 | 30 |
| CTC/AED, Zipformer-XL | 4 | 30 |
| Pruned transducer, Zipformer-S | 2 | 30 |
| Pruned transducer, Zipformer-M | 2 | 30 |
| Pruned transducer, Zipformer-L | 2 | 30 |
| Pruned transducer, Zipformer-XL | 4 | 30 |
| *CR-CTC*, Zipformer-S | 2 | 30 |
| *CR-CTC*, Zipformer-M | 2 | 30 |
| *CR-CTC*, Zipformer-L | 2 | 30 |
| *CR-CTC*, Zipformer-XL | 4 | 30 |
| *CR-CTC*/AED, Zipformer-XL | 4 | 30 |
| Pruned transducer w/ *CR-CTC*, Zipformer-XL | 4 | 30 |

Table 11: WER (%) results of different decoding methods on LibriSpeech dataset.

| Model | Greedy search decoding | | Prefix search decoding | |
|---|---|---|---|---|
| | *test-clean* | *test-other* | *test-clean* | *test-other* |
| CTC, Zipformer-S | 2.85 | 6.91 | 2.85 | 6.89 |
| CTC, Zipformer-M | 2.51 | 6.02 | 2.52 | 6.02 |
| CTC, Zipformer-L | 2.49 | 5.7 | 2.5 | 5.72 |
| *CR-CTC*, Zipformer-S | 2.57 | 5.95 | 2.52 | 5.85 |
| *CR-CTC*, Zipformer-M | 2.12 | 4.62 | 2.1 | 4.61 |
| *CR-CTC*, Zipformer-L | 2.03 | 4.37 | 2.02 | 4.35 |

Table 12: WER (%) results of different decoding methods on Aishell-1 dataset.

| Model | Greedy search decoding | | Prefix search decoding | |
|---|---|---|---|---|
| | *dev* | *test* | *dev* | *test* |
| CTC, Zipformer-S | 4.88 | 5.26 | 4.89 | 5.26 |
| CTC, Zipformer-M | 4.46 | 4.8 | 4.47 | 4.8 |
| *CR-CTC*, Zipformer-S | 3.9 | 4.12 | 3.9 | 4.12 |
| *CR-CTC*, Zipformer-M | 3.73 | 4.02 | 3.72 | 4.02 |

Table 13: WER (%) results of different decoding methods on GigaSpeech dataset.

| Model | Greedy search decoding | | Prefix search decoding | |
|---|---|---|---|---|
| | *dev* | *test* | *dev* | *test* |
| CTC, Zipformer-S | 12.15 | 12.03 | 12.08 | 11.95 |
| CTC, Zipformer-M | 11.3 | 11.31 | 11.23 | 11.27 |
| CTC, Zipformer-L | 11.21 | 11.19 | 11.16 | 11.16 |
| CTC, Zipformer-XL | 10.85 | 10.91 | 10.8 | 10.87 |
| *CR-CTC*, Zipformer-S | 11.85 | 11.8 | 11.68 | 11.58 |
| *CR-CTC*, Zipformer-M | 10.78 | 10.88 | 10.62 | 10.72 |
| *CR-CTC*, Zipformer-L | 10.42 | 10.56 | 10.31 | 10.41 |
| *CR-CTC*, Zipformer-XL | 10.28 | 10.41 | 10.15 | 10.28 |

Table 14: WER (%) results of different beam sizes for prefix search decoding on LibriSpeech dataset using Zipformer-M encoder.

| Method | Beam size | *test-clean* | *test-other* |
|--------|-----------|------------|------------|
| CTC | 1 | 2.52 | 6.02 |
| | 2 | 2.52 | 6.02 |
| | 4 | 2.52 | 6.02 |
| | 8 | 2.52 | 6.02 |
| *CR-CTC* | 1 | 2.1 | 4.61 |
| | 2 | 2.1 | 4.61 |
| | 4 | 2.1 | 4.61 |
| | 8 | 2.1 | 4.61 |

## A.4 MODEL CONFIGURATION OF DIFFERENT SCALES OF ZIPFORMER

Table 15 presents model configuration of different scales of Zipformer.

Table 15: Model configuration of Zipformer at four different scales.

| Scale | layer-numbers | embedding-dimensions | feed-forward-dimensions |
|-------|---------------|----------------------|-------------------------|
| S | {2,2,2,2,2,2} | {192,256,256,256,256,256} | {512,768,768,768,768,768} |
| M | {2,2,3,4,3,2} | {192,256,384,512,384,256} | {512,768,1024,1536,1024,768} |
| L | {2,2,4,5,4,2} | {192,256,512,768,512,256} | {512,768,1536,2048,1536,768} |
| XL | {2,2,4,5,4,2} | {192,384,768,1024,768,384} | {512,1024,2048,3072,2048,1024} |

## A.5 ABLATION STUDIES ON HYPER-PARAMETER TUNING

Table 16 presents results of tuning hyper-parameters, including $\alpha$ in Equation 3 and the ratio used to increase the amount of time masking for *CR-CTC*.

Table 16: Results of tuning $\alpha$ that controls $\mathcal{L}_{\text{CR}}$ (Equation 3) and the ratio used to increase the amount of time-masking for *CR-CTC* on LibriSpeech dataset using Zipformer-M encoder and greedy search decoding.

| Hyper-parameter | WER (%) | |
|-----------------|---------|---|
| | *test-clean* | *test-other* |
| $\alpha = 0.1$ | 2.19 | 4.8 |
| $\alpha = 0.2$ (**final**) | **2.12** | **4.62** |
| $\alpha = 0.3$ | 2.23 | 4.84 |
| $1.0\times$ time masking | 2.19 | 4.98 |
| $1.5\times$ time masking | 2.19 | 4.73 |
| $2.0\times$ time masking | 2.17 | 4.71 |
| $2.5\times$ time masking (**final**) | **2.12** | **4.62** |
| $3.0\times$ time masking | 2.17 | 4.81 |

## A.6 EMA-DISTILLED CTC

In EMA-distilled CTC, the teacher model $f^{(e)}$ is dynamically constructed for self-distillation. Its weights $\theta^{(e)}$ are updated using the exponential moving average of the current model's weights $\theta$: $\theta^{(e)} \leftarrow \tau\theta^{(e)} + (1 - \tau)\theta$, where $\tau = \min(0.9999, 1 - 10/\max(20, \text{step}))$. The teacher model $f^{(e)}$ processes the unmasked input $\mathbf{x}^{(e)}$, and produces the CTC distribution $\mathbf{z}^{(e)} = f^{(e)}(\mathbf{x}^{(e)})$ which serves as distillation target for the current model $f$. Similar to $\mathcal{L}_{\text{CR}}$ in *CR-CTC* (Equation 4), the

distillation loss $\mathcal{L}_{\mathrm{EMA}}$ is defined as:

$$\mathcal{L}_{\mathrm{EMA}}(\mathbf{z}, \mathbf{z}^{(e)}) = \sum_{t=1}^{T} D_{\mathrm{KL}}(sg(z_t^{(e)}) \| z_t). \tag{7}$$

The overall loss of EMA-distilled CTC is formulated as:

$$\mathcal{L}'' = \mathcal{L}_{\mathrm{CTC}}(\mathbf{z}, \mathbf{y}) + \gamma \mathcal{L}_{\mathrm{EMA}}(\mathbf{z}, \mathbf{z}^{(e)}), \tag{8}$$

where $\gamma$ is a hyper-parameter. In this work, we use $\gamma = 0.2$. Table 4 presents the experimental result.

## A.7 RESULTS USING CONFORMER ENCODER

To validate the effectiveness and generalization ability of our proposed *CR-CTC*, we conduct experiments on LibriSpeech dataset using a Conformer (Gulati et al., 2020) encoder, comparing different methods including CTC (Graves et al., 2006), CTC/AED (Watanabe et al., 2017), pruned transducer (Kuang et al., 2022) and *CR-CTC*. Specifically, we use a 12-layer Conformer, with an embedding dimension of 512, a convolution kernel size of 31, and a feedforward hidden dimension of 2048. For the CTC/AED model, the AED decoder is modeled by a 6-layer Transformer, where each layer has an attention dimension of 512 and a feedforward hidden dimension of 2048. The vanilla CTC model is trained for 100 epochs, while the other three models are trained for 50 epochs. Table 17 presents the experimental results. *CR-CTC* substantially outperforms the vanilla CTC and achieves marginally better results compared to the pruned transducer and CTC/AED.

Table 17: WER(%) performance of difference methods on LibriSpeech dataset using a 12-layer Conformer encoder.

| Method | Params (M) | WER (%) | |
|---|---|---|---|
| | | *test-clean* | *test-other* |
| CTC | 77.4 | 2.92 | 7.15 |
| CTC/AED | 103.1 | 2.5 | 5.94 |
| Pruned transducer | 78.6 | 2.49 | 5.87 |
| *CR-CTC* (ours) | 77.4 | **2.43** | **5.78** |

