# OpenReview forum: "CR-CTC: Consistency regularization on CTC for improved speech recognition"
_ICLR.cc/2025/Conference — ICLR 2025 Poster_

### Official Review · Reviewer_LPhF · 2024-10-31

**Soundness:** 3
**Presentation:** 3
**Contribution:** 3
**Rating:** 8
**Confidence:** 4

**Summary:**

This paper proposes a method to improve CTC performance by applying self-distillation between sub-models using drop-based techniques. The approach aims to enhance target token distribution predictions within time-masked regions and develop contextual representations from unmasked segments, drawing inspiration from self-supervised learning methods. By increasing time-masking, this technique promotes effective masked prediction, reducing peaky CTC distributions and strengthening the model's generalization ability. Experiments on multiple datasets—Librispeech, Aishell-I, and GigaSpeech—demonstrate that the proposed method achieves performance on par with transducer and CTC/AED models when used in joint training.

**Strengths:**

* This paper presents a simple yet effective distillation method by extending time-masking to develop contextual representations from unmasked segments across two different augmented views.
* It demonstrates that CTC performance is comparable to that of transducer and CTC/AED models.

**Weaknesses:**

The method generates two different augmented views by independently applying existing SpecAugment to Zipformer. However, it raises the question of how generalizable this claim is when applied to other architectures like Conformer, E-Branchformer, or Branchformer.

**Questions:**

* Why was the choice of alpha set to 0.2 in Equation 3? It would benefit readers if the authors could provide results from an ablation study showing the impact of different alpha values on performance. This would offer greater insight into the method's sensitivity to this hyperparameter choice.
* Why was a beam size of 4 specifically chosen for comparisons with other state-of-the-art models? The authors may consider including results with different beam sizes (e.g., 1, 4, 8) in an appendix to show the method's sensitivity to this parameter.
* The authors employed a larger amount of time-masking by increasing both the number of time-masking regions and the maximum masking fraction by a factor of 2.5. However, it would be interesting to know how much time-masking is optimal for masked prediction. The authors could provide results from an ablation study showing performance with different amounts of time-masking (e.g., 1x, 1.5x, 2x, 2.5x, 3x) for ZipformerXL and at least one baseline model in Table 15. This would help readers understand how critical this choice is and how generalizable the method is.
* In Tables 1, 2, and 3, are the CTC/AED baseline results reported with CTC-only decoding or CTC/AED joint decoding? This could be clarified by specifying the decoding methods for the baselines and the proposed method in the table captions.
* For a deeper understanding, the authors could include results showing the impact of increased time-masking on the baseline CTC model as well. This would help isolate whether the benefit comes from the two-branch architecture of CR-CTC or simply from more aggressive augmentation. Although the authors reported one of the baselines with larger time-masking, it would be helpful if results were provided for the other tables as well.
* In Table 3, the best results were obtained using Zipformer XL. However, the authors should:
1. Explain the rationale for using Zipformer-M in Tables 4 and 5 instead of Zipformer-XL.
2. Provide results for Zipformer-XL in Table 11 for completeness.
3. Clarify whether the results in Table 11 are from self-distillation or masked prediction in CR-CTC.
* The results of SR-CTC in Table 6 are slightly worse than those of CR-CTC. Do the authors have any explanation for this behavior, and does SR-CTC also use increased time-masking?

---

> ### Author Response · Authors · 2024-11-24
> **Thanks and response to concerns (Part 1)**
>
> We sincerely thank the reviewer for the detailed review and valuable comments, which have helped improve the clarity and quality of our work. Below, we provide detailed responses to each of the reviewer's concerns.
>
> > The method generates two different augmented views by independently applying existing SpecAugment to Zipformer. However, it raises the question of how generalizable this claim is when applied to other architectures like Conformer, E-Branchformer, or Branchformer.
>
> **In response to the comment, we have conducted additional experiments on LibriSpeech dataset using a 12-layer Conformer encoder, to validate the effectiveness and generalization ability of CR-CTC.** We compare CR-CTC with standard CTC, CTC/AED, and pruned-transducer. We train the CTC model for 100 epochs, and the other three models with 50 epochs. The following experimental results demonstrate that it is also effective with Conformer, significantly surpassing standard CTC and achieving slightly better results compared to CTC/AED and pruned transducer.
>
> - CTC, 77.4M, 2.92/7.15
> - CTC/AED, 103.1M, 2.5/5.94
> - Pruned transducer, 78.6M, 2.49/5.87
> - CR-CTC, 77.4M, 2.43/5.78
>
> **We have supplement these results in Appendix A.7 of the revised version of manuscript.**
>
> > Why was the choice of alpha set to 0.2 in Equation 3? It would benefit readers if the authors could provide results from an ablation study showing the impact of different alpha values on performance. This would offer greater insight into the method's sensitivity to this hyperparameter choice.
>
> Thanks for the comment. In the original version of manuscript, we presented the results of using different values for alpha (0.1, 0.2, 0.3) in Appendix A.5, Table 15 (Now Table 16). Setting alpha to 0.2 yielded the best result. We also mentioned this in the first paragraph of Section 4.3.
>
> > Why was a beam size of 4 specifically chosen for comparisons with other state-of-the-art models? The authors may consider including results with different beam sizes (e.g., 1, 4, 8) in an appendix to show the method's sensitivity to this parameter.
>
> **Our experimental results show that the performance of CTC models (both CR-CTC and standard CTC) is not sensitive to the beam size for prefix-search-decoding:**
>
> CTC baseline prefix-search-decoding:
> - beam-size=1: 2.52/6.02
> - beam-size=2: 2.52/6.02
> - beam-size=4: 2.52/6.02
> - beam-size=8: 2.52/6.02
>
> CR-CTC with prefix-search-decoding:
> - beam-size=1: 2.1/4.61
> - beam-size=2: 2.1/4.61
> - beam-size=4: 2.1/4.61
> - beam-size=8: 2.1/4.61
>
> **Thanks for the suggestion. We have added the results in Appendix Section A.3, Table 14, in the revised manuscript.**
>
> > The authors employed a larger amount of time-masking by increasing both the number of time-masking regions and the maximum masking fraction by a factor of 2.5. However, it would be interesting to know how much time-masking is optimal for masked prediction. The authors could provide results from an ablation study showing performance with different amounts of time-masking (e.g., 1x, 1.5x, 2x, 2.5x, 3x) for ZipformerXL and at least one baseline model in Table 15. This would help readers understand how critical this choice is and how generalizable the method is.
>
> **In the original manuscript, we presented results of different amount of time masking (ratio=1, 1.5, 2.0, 2.5, 3) for CR-CTC in Table 15 (now Table 16). The ratio of 2.5 got the best result.**
>
> Our ablation studies were all conducted with Zipformer-M encoder. We did not conduct ablation studies with the 286M-parameter Zipformer-XL on the 1k-hour LibriSpeech dataset, due to the risk of overfitting associated with training such a large model on this dataset. We provide a more detailed explanation in the latter response.
>
> > In Tables 1, 2, and 3, are the CTC/AED baseline results reported with CTC-only decoding or CTC/AED joint decoding? This could be clarified by specifying the decoding methods for the baselines and the proposed method in the table captions.
>
> Sorry for the omitted declaration of the decoding method for CTC/AED. **For CTC/AED systems, we used joint decoding that combines CTC scores and AED scores [1].** This information has been added to Section 4.1 (Implementation Details) in the revised version of the manuscript.
>
> [1] Watanabe, Shinji, et al. "Hybrid CTC/attention architecture for end-to-end speech recognition." IEEE Journal of Selected Topics in Signal Processing 11.8 (2017): 1240-1253.

---

> ### Author Response · Authors · 2024-11-24
> **Thanks and response to concerns (Part 2)**
>
> > For a deeper understanding, the authors could include results showing the impact of increased time-masking on the baseline CTC model as well. This would help isolate whether the benefit comes from the two-branch architecture of CR-CTC or simply from more aggressive augmentation. Although the authors reported one of the baselines with larger time-masking, it would be helpful if results were provided for the other tables as well.
>
> In the original manuscript, Table 5, we provided the result of using a larger amount of time masking (ratio = 2.5) for the CTC baseline model, which led to worse result. In addition, we reported the result of using larger frequency masking for CR-CTC, which led to a WER degradation of 0.07% on test-clean. This indicates that the performance gain from increasing the amount of time masking is primarily due to the masked prediction behavior, rather than merely increasing the input diversity for the two branches, or the more aggressive augmentation.
>
> We conducted this ablation study on LibriSpeech, consistent with our other ablation studies.
>
> In response to the comment, we conducted an additional experiment using larger amount of time masking for the CTC baseline on Aishell-1 dataset, and the results also showed a performance degradation:
> - CTC: 4.47, 4.8
> - CTC, larger time masking: 4.49, 4.86
>
> > In Table 3, the best results were obtained using Zipformer XL. However, the authors should: 1. Explain the rationale for using Zipformer-M in Tables 4 and 5 instead of Zipformer-XL. 2. Provide results for Zipformer-XL in Table 11 for completeness. 3. Clarify whether the results in Table 11 are from self-distillation or masked prediction in CR-CTC.
>
> In this work, our ablation studies (e.g., Tables 4, 5, 6, and 7) were all conducted using Zipformer-M encoder on the 1k-hour LibriSpeech dataset. The model size is moderate and the dataset is widely used. This is also consistent with the Zipformer paper [1]. We believe our choice is appropriate and not computation-costly for ablation studies.
>
> In response to the comment, we have conducted experiments to train Zipformer-XL with CTC and CR-CTC, respectively. The results conform that CR-CTC significantly outperforms the CTC baseline:
> - CTC, 2.35/5.45
> - CR-CTC, 2.02/4.34
>
> However, the results reported in the literature for the LibriSpeech dataset, such as those from Zipformer [1], Branchformer[2], E-Branchformer[3], and Conformer [4], typically use models with a maximum size of no more than 200M parameters. For example, in the Zipformer paper, the largest model reported for LibriSpeech dataset is Zipformer-L. We believe this limitation is due to the risk of overfitting when training larger models on the 1k-hour LibriSpeech dataset. In our experience, we tried training a Zipformer-XL with the pruned transducer loss, but it resulted in overfitting and performed worse than Zipformer-L. Therefore, we did not include Zipformer-XL in the LibriSpeech table. This choice was also made to ensure a fair comparison in terms of model size with other state-of-the-art models.
>
> Tables 4, 5, and 6 presents the ablation study results for self-distillation, masked prediction, and peak suppression, which explain the three perspectives for CR-CTC. We have clarified this in the table captions. All other tables present the results with our final CR-CTC model, unless otherwise specified.
>
> [1] Yao, Zengwei, et al. "Zipformer: A faster and better encoder for automatic speech recognition." The Twelfth International Conference on Learning Representations. 2024.
>
> [2] Peng, Yifan, et al. "Branchformer: Parallel mlp-attention architectures to capture local and global context for speech recognition and understanding." International Conference on Machine Learning. PMLR, 2022.
>
> [3] Kim, Kwangyoun, et al. "E-branchformer: Branchformer with enhanced merging for speech recognition." 2022 IEEE Spoken Language Technology Workshop (SLT). IEEE, 2023.
>
> [4] Gulati, Anmol, et al. "Conformer: Convolution-augmented transformer for speech recognition." arXiv preprint arXiv:2005.08100 (2020).
>
> > The results of SR-CTC in Table 6 are slightly worse than those of CR-CTC. Do the authors have any explanation for this behavior, and does SR-CTC also use increased time-masking?
>
> In this work, we provide three explanation perspectives for the essential behaviors in CR-CTC: self-distilltion, masked prediction, and peak suppression. Inspired by the point of peak suppression, we additionally propose the simple method, SR-CTC, specifically designed to learn smoother CTC distributions (Appendix Section A.1), which is experimentally validated to be effective (Table 6). The reason that it is worse than our final CR-CTC is as expected, since it does not benefit from the other two key behaviors: self-distilltion, masked prediction. Unlike CR-CTC, SR-CTC does not leverage the token distributions from another branch, thereby lacking the masked prediction behavior. So we didn't use larger time masking for SR-CTC.

---

> > ### Comment · Reviewer_LPhF · 2024-11-26
> > **Response to Authors**
> >
> > Thank you for addressing my comments and providing additional analysis. I believe these improvements have enhanced the overall presentation of the paper. I am satisfied with the revisions and will maintain my score.

---

> > > ### Author Response · Authors · 2024-11-26
> > > **Thanks for feedback**
> > >
> > > We’re happy that our response addressed your concerns. Thank you for your feedback!

---

### Official Review · Reviewer_HiGE · 2024-10-31

**Soundness:** 3
**Presentation:** 3
**Contribution:** 3
**Rating:** 8
**Confidence:** 5

**Summary:**

Consistency regularization (CR) is a well established existing method, where you forward through some model two times with different augmentation (and maybe also dropout or other randomness) to get two predictions, and then you minimize the symmetric KL between both, or similar. Thus, this method is purely on the training side, and doesn't change any modeling aspect.

Here, CR is applied to speech recognition, specifically to CTC models, on a frame-by-frame basis, called CR-CTC. The main difference in the two branches is caused by different SpecAugment masking.

For fair comparison, due to forwarding the data twice now, the number of epochs and the batch size are both halfed for the CR-CTC case.

Experiments are done on Librispeech, Gigaspeech and Aishell.

The method is mainly tested on CTC, but then some extension to that is when they used a joint AED/CTC model in the end, where CR is applied only to the CTC part.

A number of ablations has been made on the loss scale and on increasing the SpecAugment masking, which seems ot help more on CR-CTC, but for pure CTC, the original amount of SpecAugment masking already seems optimal.

The improvements on Librispeech test-other are quite large: From 5.72% WER to 4.35% WER.

It is also shown that it reduces the peakiness of the alignment behavior a bit.

**Strengths:**

- Simple method.

- Seems to give huge improvements, at least in some settings.

**Weaknesses:**

- Some smaller details are a bit unclear.
- Only tested for Zipformer. More standard would be Conformer, but this is missing.
- Unclear how well this method works in other cases, e.g. other models, other datasets, some other hyperparams different. Specifically, I tested it in my setup, and it didn't really helped.

**Questions:**

Abstract starts a bit strange. It says CTC is worse than RNN-T and AED. Yes sure, we know. But then it talks about some method to maybe improve CTC. So why is mentioning RNN-T/AED relevant? Is it because you think the gap between CTC and AED/RNN-T is larger than what you would expect, and some methods like the presented here should close the gap? But I don't think that this really is being shown here in this work. Also, some variant of this method could maybe be applied to AED/RNN-T just as well. So, I don't really see why mentioning AED/RNN-T in the abstract is really relevant for this work here. It's fine in the introduction, to put CTC into perspective, but I don't think it's relevant in the abstract. I was a bit confused about this.

Eq 3 and also Figure 1, the CTC loss is maybe better formulated on z, not on x? I found it weird that x goes into L_CTC but z goes into L_CR.


"a time warping factor of 80" - what does that mean? I don't think you make the sequence 80 times longer?

Please clarify the downsampling of the Zipformer. Do you stick to the original Zipformer here, where the Conv Frontend downsamples the 100Hz feature frames to 50Hz, and then the residual/bypass connection is always at 50Hz, and at the very end, you downsample again to get 25Hz output frames, i.e. the log probs are at 25 Hz?

Did you investigate how the downsampling influences the CR-CTC loss? I think this can have quite a crucial impact, as we know that in general, for CTC, the ratio of input length to output length plays an important role for convergence and training dynamics.

Did you also test with other vocab sizes? 500 BPE size seems quite small to me. How does it perform with larger vocab sizes, e.g. with 10k?

How does the Zipformer influence the results? Specifically, do you think you get the same improvements with a normal Conformer?


"auxiliary head of AED" (p9, l483) (and also same with transducer) / Table 7: I don't exactly understand what you report here. Is the AED (or transducer) head just used as an aux loss, and during recognition, you only use the CTC head and ignore the AED (or transducer)? Please be more clear about that. Also, you are giving the wrong citation for that. The reference you give is about joint AED/CTC, where both AED and CTC heads are used for recognition, so nothing is ignored, nothing is just used as aux loss. The only reference I know where AED is used as an aux loss for CTC is "Keep Decoding Parallel With Effective Knowledge Distillation From Language Models To End-To-End Speech Recognisers", Hentschel et al, 2024.


On GigaSpeech, improvement seems much less (XL, test: 10.87 -> 10.28) compared to Librispeech (5.72 -> 4.35). Why?

Table 3 caption: "GigaSpeeech" typo.


Transducer w/ CR-CTC, what exactly is that? The same approach applied on transducer? But then this is not CTC? Or is it combined CTC with transducer?


Note, as your method is very simple to implement, and your improvements here are really impressive, I was just trying it out myself. However, with negative result: For my Conformer CTC baseline, on 100Hz inputs, downsampled by factor 6, with BPE 10k vocab, with aux AED loss ("Keep Decoding Parallel With Effective Knowledge Distillation From Language Models To End-To-End Speech Recognisers", Hentschel et al, 2024), where my baseline with greedy decoding without LM was at 5.93% on dev-other, it degraded with CR-CTC to 5.99% on dev-other. I halved the number epochs and halved the batch size for the CR experiment, just like you did. This is with CR loss scale 0.2. I did not adapt SpecAugment yet, but from your paper, I would expect that even with this setting, I should already see quite some improvement. So, why don't I? Your paper is lacking such study on other settings, as mentioned above (Conformer, other BPE sizes, other downsampling) to know whether I can/should expect similar improvements there or not, and whether I maybe need a very different CR loss scale there, or whether I need to care about other things.


Note, halving the batch size can have other effects. Many methods (e.g. optimizer, LR schedule, regularization, etc) don't work in the same way for different batch sizes. You do effectively more updates to the model. It can also have a regularization effect. So, I think an important missing experiment is: What happens to the baseline when you half the batch size? Maybe you also get improvements there?

**Details Of Ethics Concerns:**

x

---

> ### Author Response · Authors · 2024-11-24
> **Thanks and response to concerns (Part 1)**
>
> We sincerely thank the reviewer for the detailed review and valuable comments, which have helped improve the clarity and quality of our work. Below, we provide detailed responses to each of the reviewer's concerns.
>
> > Some smaller details are a bit unclear.
>
> We have updated the manuscript to improve its clarity. Please see the responses below.
>
> > Abstract starts a bit strange. It says CTC is worse than RNN-T and AED. Yes sure, we know. But then it talks about some method to maybe improve CTC. So why is mentioning RNN-T/AED relevant? Is it because you think the gap between CTC and AED/RNN-T is larger than what you would expect, and some methods like the presented here should close the gap? But I don't think that this really is being shown here in this work. Also, some variant of this method could maybe be applied to AED/RNN-T just as well. So, I don't really see why mentioning AED/RNN-T in the abstract is really relevant for this work here. It's fine in the introduction, to put CTC into perspective, but I don't think it's relevant in the abstract. I was a bit confused about this.
>
> Thanks for pointing this out. Our main goal is indeed to improve CTC performance, and the results demonstrate that our proposed CR-CTC achieves state-of-the-art results comparable to those attained by transducer and CTC/AED. In the revised manuscript, we have refined the abstract to make the description more clear.
>
> > Eq 3 and also Figure 1, the CTC loss is maybe better formulated on z, not on x? I found it weird that x goes into L_CTC but z goes into L_CR.
>
> Thanks! We have replaced x with z in L_CTC, in the revised manuscript.
>
> > "a time warping factor of 80" - what does that mean? I don't think you make the sequence 80 times longer?
>
> Sorry for making that confusing. As described in the Section 4.1, we use Lhotse (https://github.com/lhotse-speech/lhotse) for data preparation. "time_warp_factor" is a parameter of the "time_warp" function (https://github.com/lhotse-speech/lhotse/blob/master/lhotse/dataset/signal_transforms.py#L338). Specifically, it specifies the maximum range (in frames) around a randomly selected center point on the time axis where the warping can occur. The "warped" index is chosen randomly within the range [center - factor, center + factor]. Then it interpolates the fisrt "center" frames to "warped" frames (denoted as A), and interpolates the remaining "T - center" frames to "T - warped" frames (denoted as B), where T is the length of input length. The obtained two parts A and B are concatenated as the result. We have added a footnote with the link to make it more clear in the revised manuscript.
>
> > Please clarify the downsampling of the Zipformer. Do you stick to the original Zipformer here, where the Conv Frontend downsamples the 100Hz feature frames to 50Hz, and then the residual/bypass connection is always at 50Hz, and at the very end, you downsample again to get 25Hz output frames, i.e. the log probs are at 25 Hz?
>
> Yes. We use the original downsampling rates of Zipformer. It takes input features at frame rate of 100Hz, processes the sequence through 6 stacks with frame rates of 50Hz, 25Hz, 12.5Hz, 6.25Hz, 12.5Hz, and 25Hz, and finally produces the encoder output at frame rate of 25Hz.
>
> Thanks for your suggestion. We have added this information in the revised manuscript.
>
> > "auxiliary head of AED" (p9, l483) (and also same with transducer) / Table 7: I don't exactly understand what you report here. Is the AED (or transducer) head just used as an aux loss, and during recognition, you only use the CTC head and ignore the AED (or transducer)? Please be more clear about that. Also, you are giving the wrong citation for that. The reference you give is about joint AED/CTC, where both AED and CTC heads are used for recognition, so nothing is ignored, nothing is just used as aux loss. The only reference I know where AED is used as an aux loss for CTC is "Keep Decoding Parallel With Effective Knowledge Distillation From Language Models To End-To-End Speech Recognisers", Hentschel et al, 2024.
>
> Thanks for the suggestion. Yes, in Table 7, the AED and transducer heads are discarded after training, with only the CTC head retained for inference. In the revised manuscript, we have clarified this point for improved clarity and updated the reference with the one you suggested.
>
> > Table 3 caption: "GigaSpeeech" typo.
>
> Thanks. We have corrected it in the revised manuscript.
>
> > Transducer w/ CR-CTC, what exactly is that? The same approach applied on transducer? But then this is not CTC? Or is it combined CTC with transducer?
>
> As described in the first paragraph of Section 4.2, "Pruned transducer w/ CR-CTC" refers to using CR-CTC as an auxiliary loss to improve the pruned transducer model. The system consists of a CTC head and a transducer head, with consistency regularization only applied on the CTC head (i.e., CR-CTC). After training, the CTC head is discarded, retaining only the transducer head for inference.

---

> ### Author Response · Authors · 2024-11-24
> **Thanks and response to concerns (Part 2)**
>
> > Only tested for Zipformer. More standard would be Conformer, but this is missing.
>
> > How does the Zipformer influence the results? Specifically, do you think you get the same improvements with a normal Conformer?
>
> **In response to the comments, we have conducted additional experiments on LibriSpeech dataset using a 12-layer Conformer encoder, to validate the effectiveness and generalization ability of CR-CTC.** We compare CR-CTC with standard CTC, CTC/AED, and pruned-transducer. We train the CTC model for 100 epochs, and the other three models with 50 epochs. **The following experimental results demonstrate that it is also effective with Conformer, significantly surpassing standard CTC and achieving slightly better results compared to CTC/AED and pruned transducer.**
>
> - CTC, 77.4M, 2.92/7.15
> - CTC/AED, 103.1M, 2.5/5.94
> - Pruned transducer, 78.6M, 2.49/5.87
> - CR-CTC, 77.4M, 2.43/5.78
>
> **We have supplement these results in Appendix A.7 of the revised version of manuscript.**
>
> > Did you investigate how the downsampling influences the CR-CTC loss? I think this can have quite a crucial impact, as we know that in general, for CTC, the ratio of input length to output length plays an important role for convergence and training dynamics.
>
> In our paper, we adopt the commonly used downsampling rate of 4, where the input frame rate is 100 Hz, and the output frame rate is 25 Hz.
>
> In response to the comment, we have conducted additional experiments on LibriSpeech dataset to investigate the impact of **different encoder downsampling rates (2, 4, 8)** on CTC and CR-CTC, with Zipformer-M encoder, by changing the downsampling rates in the output Downsample module in Zipformer. **The following experimental results on test-clean/test-other (WER %) with greedy-search-decoding demonstrate that CR-CTC consistently outperforms CTC baseline across different downsampling rates.** Interestingly, increasing the downsampling rate from 4 to 8 slightly improves the CTC baseline, though its performance remains notably inferior to that of CR-CTC.
> - Downsampling rate = 2:
>   - CTC, train for 100 epochs, 3.25/7.91;
>   - CR-CTC, train for 50 epochs, 2.38/5.36
> - Downsampling rate = 4 (current setting):
>   - CTC, train for 100 epochs, 2.51/6.02;
>   - CR-CTC, train for 50 epochs, 2.12/4.62
> - Downsampling rate = 8:
>   - CTC, train for 100 epochs, 2.44/5.67;
>   - CR-CTC, train for 50 epochs, 2.12/4.74
>
> > Did you also test with other vocab sizes? 500 BPE size seems quite small to me. How does it perform with larger vocab sizes, e.g. with 10k?
>
> As described in the manuscript, our experiments are conducted using the Icefall framework, where the default BPE size is set to 500 for both the LibriSpeech and GigaSpeech datasets.
>
> **In response to the comment, we have conducted additional experiments on LibriSpeech dataset to test using a large vocab size of 10k with Zipformer-M encoder.** Experimental results show that increasing the vocab size from 500 to 10k leads to performance degradation for both CTC and CR-CTC. This indicates that the vocab size of 10k might be too large for the 1k-hour LibriSpeech dataset. **It is worth mentioning that CR-CTC still significanly outperforms CTC baseline with the vocab size of 10k.**
> - BPE vocab size = 500 (current setting):
>   - CTC, train for 50 epochs, 2.77/6.6
>   - CR-CTC, train for 25 epochs, 2.3/5.23
> - BPE vocab size = 10k:
>   - CTC, train for 50 epochs, 3.03/6.65
>   - CR-CTC, train for 25 epochs, 2.46/5.46
>
> > Note, halving the batch size can have other effects. Many methods (e.g. optimizer, LR schedule, regularization, etc) don't work in the same way for different batch sizes. You do effectively more updates to the model. It can also have a regularization effect. So, I think an important missing experiment is: What happens to the baseline when you half the batch size? Maybe you also get improvements there?
>
> As CR-CTC requires two forward pass during training, we train CR-CTC models with half the batch size and half the number of epochs compared to CTC models, ensuring a fair comparison in terms of training cost. **The total number of iterations remains the same for both models.**
>
> **In response to the comment, we have experimented with using half the batch size and half the number of epochs for the CTC baseline.** Similar to CR-CTC, this adjustment resulted in each epoch having twice the number of iterations. **However, our results indicate that this leads to performance degradation for the CTC baseline model.**
> - CTC Baseline: train for 100 epochs, 2.51/6.02
> - CTC, train for 50 epochs, half batch size, 2.76/6.5
> - CR-CTC, train for 50 epochs, half batch size, 2.12/4.62

---

> ### Author Response · Authors · 2024-11-24
> **Thanks and response to concerns (Part 3)**
>
> > Unclear how well this method works in other cases, e.g. other models, other datasets, some other hyperparams different. Specifically, I tested it in my setup, and it didn't really helped.
>
> > Note, as your method is very simple to implement, and your improvements here are really impressive, I was just trying it out myself. However, with negative result: For my Conformer CTC baseline, on 100Hz inputs, downsampled by factor 6, with BPE 10k vocab, with aux AED loss ("Keep Decoding Parallel With Effective Knowledge Distillation From Language Models To End-To-End Speech Recognisers", Hentschel et al, 2024), where my baseline with greedy decoding without LM was at 5.93% on dev-other, it degraded with CR-CTC to 5.99% on dev-other. I halved the number epochs and halved the batch size for the CR experiment, just like you did. This is with CR loss scale 0.2. I did not adapt SpecAugment yet, but from your paper, I would expect that even with this setting, I should already see quite some improvement. So, why don't I? Your paper is lacking such study on other settings, as mentioned above (Conformer, other BPE sizes, other downsampling) to know whether I can/should expect similar improvements there or not, and whether I maybe need a very different CR loss scale there, or whether I need to care about other things.
>
> **In response to the comments, as mentioned above, we have conducted additional experiments to evaluate the use of a Conformer encoder, a larger BPE vocabulary size of 10k, and alternative encoder downsampling rates of 2 and 8. The results from these experiments consistently demonstrate that CR-CTC significantly improves CTC performance.**
>
> Concerning the lack of improvement with CR-CTC in your system, **I suspect the issue might be caused an incorrect relative scales when summing up different losses.** This could be related to how the **"reduction"** parameter is specified for the batch of loss values. For example, the reduction in our PyTorch-based code is as follows:
>
> ```python
> # ctc_output: (2 * batch_size, seq_len, vocab_size), the log-probs
> # ctc_output_lens: (2 * batch_size,)
> # targets: (sum(target_lengths))
> # target_lengths: (2 * batch_size,)
>
> # Compute CTC loss
> ctc_loss = torch.nn.functional.ctc_loss(
>     log_probs=ctc_output.permute(1, 0, 2),
>     targets=targets.cpu(),
>     input_lengths=ctc_output_lens.cpu(),
>     target_lengths=target_lengths.cpu(),
>     reduction="sum",
> )
>
> # Compute CR loss
> cr_targets = ctc_output.detach().chunk(2, dim=0)  # stop-grad
> cr_targets = torch.cat([cr_targets[1], cr_targets[0]], dim=0)  # exchange
> cr_loss = nn.functional.kl_div(
>     input=ctc_output,
>     target=cr_targets,
>     reduction="none",
>     log_target=True,
> )
> length_mask = pad_mask(ctc_output_lens).unsqueeze(-1)  # True for padding positions
> cr_loss = cr_loss.masked_fill(length_mask, 0.0).sum()
>
> # The following lines are optional if we are using Adam-like optimizer (which is invariant to gradient scale)
> # Scale ctc_loss and cr_loss by the total number of frames
> tot_frames = ctc_output_lens.sum().item()
> ctc_loss = ctc_loss / tot_frames
> cr_loss = cr_loss / tot_frames
> ```
> In compliance with the anonymous review policy, the link to our complete code will be included in the final version of the paper. Perhaps you could refer to it then.
>
> If your system is a hybrid CTC/AED, another potential issue may arise from how the loss scales for CR loss, CTC loss, and AED loss are specified. **It is important to maintain the relative scale between the CR loss scale and CTC loss scale, for example, keeping it at 0.2.** If your original loss weights were 0.1 for the CTC loss and 0.9 for the AED loss, then with the CR loss, the new scaling would be 0.02 for the CR loss, 0.1 for the CTC loss, and 0.9 for the AED loss.

---

> ### Author Response · Authors · 2024-11-24
> **Thanks and response to concerns (Part 4)**
>
> > On GigaSpeech, improvement seems much less (XL, test: 10.87 -> 10.28) compared to Librispeech (5.72 -> 4.35). Why?
>
> Thanks for the comment. We agree that the performance gain of CR-CTC on the 10k-hour GigaSpeech dataset is smaller compared to the 1k-hour LibriSpeech dataset. **This is consistent with our expectation, as a regularization method typically provides smaller performance gain with larger training datasets due to reduced overfitting.** However, we would like to emphasize the following results on GigaSpeech dataset:
> - **CR-CTC still significantly improves the performance of CTC models**
> - **CR-CTC achieves performance comparable to CTC/AED and pruned transducer models with Zipformer-L/XL encoders**
> - **Using CR-CTC for joint training can further enhance the performance of both CTC/AED and pruned transducer models.**
>
> **To validate the effectiveness and generalization ability of CR-CTC with a large amount of training data, we additionally train the Zipformer-XL model with CTC and CR-CTC, seperately, on a 50k-hour English dataset, LibriHeavy (https://github.com/k2-fsa/libriheavy), and decode on LibriSpeech test sets.** (Specifically, in line with all experiments in the maniscript, as CR-CTC involves two model forward pass, we train the CR-CTC model with half the batch size and half the number of epochs compared to the CTC model, ensuring a fair comparison in terms of training cost.) **Experimental results (WER %) on LibriSpeech test-clean/test-other demonstrate that it can still signficicantly improve the CTC performance:**
> - CTC, train for 12 epochs, greedy-search-decoding: 2.14/4.65; prefix-search-decoding: 2.14/4.66
> - CR-CTC, train for 6 epochs, greedy-search-decoding: 1.94/3.57; prefix-search-decoding: 1.92/3.58

---

### Official Review · Reviewer_6WyR · 2024-11-03

**Soundness:** 3
**Presentation:** 3
**Contribution:** 2
**Rating:** 5
**Confidence:** 4

**Summary:**

This paper proposes an improved version of Connectionist Temporal Classification (CTC) called Consistency-Regularized CTC (CR-CTC) for automatic speech recognition (ASR). CR-CTC enforces consistency between two CTC distributions obtained from different augmented views of the input speech mel-spectrogram.  The proposed method has 3 advantages: 1) it conducts self-distillation between random pairs of sub-models that process different augmented views; 2) it learns contextual representation through masked prediction for positions within time-masked regions, especially when we increase the amount of time masking; 3) it suppresses the extremely peaky CTC distributions, thereby reducing overfitting and improving the generalization ability. Extensive experiments on LibriSpeech, Aishell-1, and GigaSpeech datasets demonstrate the effectiveness of CR-CTC, which achieves performance comparable to, or even slightly better than, that of transducer and CTC/AED.

**Strengths:**

CR-CTC takes two different augmented views of the same speech mel-spectrogram as the inputs and enforce consistency between the two obtained CTC distributions. This method helps the model to do self-distillation between randomly sampled sub-models, learn contextual representation through masked prediction and reduce the peaky CTC distribution. The idea is simple and easy to implement. The training cost didn’t increase based on the description in section 4.1. The experiments on LibriSpeech, Aishell-1, and GigaSpeech show the proposed method outperform standard CTC in all these sets for different model sizes. It achieves comparable accuracy or even better than the advanced transducer or CTC/AED model. The paper also provided detailed ablation study to help the reader understand more details about the method.

**Weaknesses:**

Comparing results in table 1 and 3, the advantages of CR-CTC over standard CTC is smaller for GigaSpeech set than that for LibriSpeech set.  This may indicate that the proposed method may not work very well for big training data, e.g. tens of thousands of speech hours.
    In table 1, 2 and 3, it’s not mentioned that the results for transducer and CTC/AED models are from beam search or greedy search. For these two models, the results of beam or greedy search usually have big differences. If the results given are from greedy search, it may mean the accuracy of CR-CTC model still have gap from that of transducer and CTC/AED model with beam search.

**Questions:**

For the combination of transducer and CR-CTC model, does the CTC score is used during the decoding?

---

> ### Author Response · Authors · 2024-11-21
> **Thanks and response to concerns**
>
> We sincerely thank the reviewer for the detailed review and valuable comments, which have helped improve the clarity and quality of our work. Below, we provide detailed responses to each of the reviewer's concerns.
>
> > Comparing results in table 1 and 3, the advantages of CR-CTC over standard CTC is smaller for GigaSpeech set than that for LibriSpeech set. This may indicate that the proposed method may not work very well for big training data, e.g. tens of thousands of speech hours.
>
> Thanks for the comment. We agree that the performance gain of CR-CTC on the 10k-hour GigaSpeech dataset is smaller compared to the 1k-hour LibriSpeech dataset. This aligns with our expectation, as regularization methods typically yield smaller gains with larger training datasets due to reduced overfitting. However, we would like to highlight the following results on GigaSpeech dataset:
> - **CR-CTC still significantly improves the WER (%) performance of CTC models:**
>   - 12.08/11.95 → 11.68/11.58 with Zipformer-S,
>   - 11.23/11.27 → 10.62/10.72 with Zipformer-M,
>   - 11.16/11.16 → 10.31/10.41 with Zipformer-L,
>   - 10.8/10.87 → 10.15/10.28 with Zipformer-XL.
> - **Compared to CTC/AED and pruned transducer models, CR-CTC achieves comparable performance on Zipformer-L/XL models.**
> - **Employing CR-CTC for joint training further improves the performance of both CTC/AED and pruned transducer models.** Specifically, with Zipformer-XL, it gets WER (%) performance improvements: 10.22/10.33 -> 9.92/10.07 for CTC/AED, 10.09/10.2 -> 9.95/10.03 for pruned transducer.
>
> > In table 1, 2 and 3, it’s not mentioned that the results for transducer and CTC/AED models are from beam search or greedy search. For these two models, the results of beam or greedy search usually have big differences. If the results given are from greedy search, it may mean the accuracy of CR-CTC model still have gap from that of transducer and CTC/AED model with beam search.
>
> Sorry for the omitted declaration of the decoding methods used in our results for the transducer and CTC/AED models. For the pruned transducer models, we did employ beam search decoding [1], while for the CTC/AED models, we did use joint decoding by combining attention-based and CTC scores [2]. **Therefore, the comparisons are fair.** Thanks for your question. This information has been added to Section 4.1 (Implementation Details) in the revised version of the manuscript.
>
> [1] Kang, Wei, et al. "Fast and parallel decoding for transducer." ICASSP 2023-2023 IEEE International Conference on Acoustics, Speech and Signal Processing (ICASSP). IEEE, 2023.
>
> [2] Watanabe, Shinji, et al. "Hybrid CTC/attention architecture for end-to-end speech recognition." IEEE Journal of Selected Topics in Signal Processing 11.8 (2017): 1240-1253.
>
> > For the combination of transducer and CR-CTC model, does the CTC score is used during the decoding?
>
> When using CR-CTC as an auxiliary loss to improve transducer models, **we only utilize the transducer head for decoding, without incorporating the CTC scores.** Thanks for your question. We have added this information in the first paragraph of Section 4.2.

---

> > ### Comment · Reviewer_6WyR · 2024-11-22
> >
> > 1 For the accuracy improvement comparison between LibriSpeech and GigaSpeech.  Yes, CR-CTC still improves the accuracy obviously compared with  CTC, but  the relative gain for LibriSpeech is larger than that for GigaSpeech (~20% vs. less than 10%).
> > 2 For the decoding method for Transducer and AED models: yes, the comparison is fair if beam search are also used for these models.

---

> > > ### Author Response · Authors · 2024-11-22
> > > **Thanks for feedback**
> > >
> > > Thank you very much for your feedback! We have supplemented additional experimental results to validate the effectiveness and generalization ability of CR-CTC on 50k-hour training data, as detailed in the comment below. Thank you again!

---

> > > ### Author Response · Authors · 2024-11-25
> > > **Thanks for feedback**
> > >
> > > Thank you very much for your feedback!
> > >
> > > First, we would like to summarize the contributions of this work:
> > > - We propose CR-CTC, which enforces consistency between two CTC distributions obtained from different augmented views of the input mel-spectrogram. We also provide in-depth insights into its essential behaviors from three perspectives: self-distillation, masked prediction, and peak suppression.
> > > - Experiments on LibriSpeech, Aishell-1, and GigaSpeech datasets domenstrate that CR-CTC significantly improves the CTC performance, achieving state-of-the-art results comparable to those attained by transducer or CTC/AED.
> > >
> > > In response to the reviews, we have also made **several updates** to the manuscipt:
> > > - We have updated the paragraphs discussing related works on consistency regularization in Section 2 to more explicitly clarify the distinctions of our work.
> > > - We have supplemented an additional self-distillation experiment, EMA-distilled CTC, in Section 4.3. Experimental results in Table 4 show that CR-CTC significantly outperforms EMA-distilled CTC.
> > > - We have conducted additional experiments on LibriSpeech dataset **using a Conformer encoder**, to validate the effectiveness and generalization ability of CR-CTC. **Results in Appendix A.7 demonstrate that it is also effective with Conformer, significantly surpassing standard CTC and achieving slightly better results compared to CTC/AED and transducer.**
> > >   - CTC, 77.4M, 2.92/7.15
> > >   - CTC/AED, 103.1M, 2.5/5.94
> > >   - Pruned transducer, 78.6M, 2.49/5.87
> > >   - CR-CTC, 77.4M, 2.43/5.78
> > > - We have also updated some details to improve clarity.
> > >
> > > Do you have any further questions? We would be happy to address any concerns you may have. If there are no other issues, would you consider updating the score?

---

> > > > ### Author Response · Authors · 2024-11-26
> > > > **Thanks for review and feedback**
> > > >
> > > > Thank you for your detailed review, valuable comments, and patient feedback. We hope our revisions have addressed your concerns. If there are any further details we can improve, please feel free to let us know.
> > > >
> > > > We sincerely appreciate your time and effort. Thanks a lot!

---

> ### Author Response · Authors · 2024-11-22
> **Additional experiment to validate the effectiveness and generalization ability of CR-CTC on 50k-hour training data**
>
> To validate the effectiveness and generalization ability of CR-CTC with a large amount of training data, we additionally train the Zipformer-XL model with CTC and CR-CTC, seperately, on a **50k-hour English dataset**, LibriHeavy (https://github.com/k2-fsa/libriheavy), and decode on LibriSpeech test sets. (Specifically, in line with all experiments in the maniscript, as CR-CTC involves two model forward pass, we train the CR-CTC model with half the batch size and half the number of epochs compared to the CTC model, ensuring a fair comparison in terms of training cost.) **Experimental results (WER %) on LibriSpeech test sets (test-clean/test-other) demonstrate that it can still significantly improve the CTC performance, when using a large amount of training data (50k hours)**:
> - CTC, train for 12 epochs, greedy-search-decoding: 2.14/4.65; prefix-search-decoding: 2.14/4.66
> - CR-CTC, train for 6 epochs, greedy-search-decoding: 1.94/3.57; prefix-search-decoding: 1.92/3.58

---

> ### Author Response · Authors · 2024-12-02
> **Additional experiment on WenetSpeech dataset to validate the effectiveness and generalization ability of CR-CTC**
>
> To validate the effectiveness and generalization ability of CR-CTC with a large amount of training data, we additionally train the Zipformer-L model with CTC and CR-CTC, seperately, on a 10k-hour Mandarin dataset, WenetSpeech (https://github.com/wenet-e2e/WenetSpeech). Experimental results (WER %) on test sets (TEST_NET/TEST_MEETING) demonstrate that CR-CTC can still significantly improve the CTC performance.
>
> * CTC, train for 18 epochs, greedy-search-decoding: 7.73/10.81 ; prefix-search-decoding: 7.73/10.83
> * CR-CTC, train for 9 epochs, greedy-search-decoding: **6.68/8.74**  ; prefix-search-decoding: **6.63/8.63**
>
> **Notably, the relative improvements on TEST_MEETING (which is out-of-domain) are around 20%, demonstrating the generalization ability of CR-CTC.** We hope this addresses your concerns and that you will reconsider our work. Thank you very much!

---

### Official Review · Reviewer_LMc2 · 2024-11-03

**Soundness:** 3
**Presentation:** 3
**Contribution:** 3
**Rating:** 6
**Confidence:** 5

**Summary:**

This paper applies a new type of self-consistency loss on different augmented view for CTC based ASR model. The new consistency regularized loss is doing KL over CTC output. Experimental results shows WER got improved.

**Strengths:**

The proposed idea is intuitive. Although the gain is small, the paper is compared with couple competitive baseline on librispeech.

**Weaknesses:**

The paper lack of citation to many very relevant work:

Most important one:
Contrastive siamese network for semi-supervised speech recognition (https://arxiv.org/pdf/2205.14054). The paper focus on compare with SoTA, instead of compare with literature self-distill or other consistency based baseline. In that paper, it include practical trick to make SimSiam type of model work for ASR.

**Questions:**

Please properly cite literature work and make proper comparison.

---

> ### Author Response · Authors · 2024-11-21
> **Thanks and response to concerns (part 1)**
>
> We sincerely thank the reviewer for the valuable and insightful comments and pointing out the missing references, which greatly contribute to improving the clarity and quality of our work. Below, we address each of the reviewer's concerns in detail.
>
> > the gain is small
>
> **We would like to argue that the performance gain achieved by CR-CTC is highly significant,** as denomstrated by the experimental results on LibriSpeech, AiShell-1, and GigaSpeech datasets (Table 4, 5, 6). (Note: our primary goal is to enhance CTC performance and narrow the performance gap between CTC and transducer or CTC/AED systems.) Below, we summarize our experimental results:
> - **CR-CTC significantly improves the performance of CTC.** For example, on LibriSpeech, CR-CTC achieves the following WER (%) improvements: 2.85/6.89 ->  2.52/5.85 with Zipformer-S, 2.52/6.02 -> 2.1/4.61 with Zipformer-M, 2.5/5.72 ->  2.02/4.35 with Zipformer-L.
> - **CR-CTC achieves results comparable to, or even slightly better than, those of transducer and CTC/AED.  It is worth mentioning that this is the first work to enable CTC models to match the performance of transducer and CTC/AED systems.**
> - CR-CTC can further improve the performance of transducer and CTC/AED when employed for jointly training, achieving **new state-of-the-art results.** For instance, on LibriSpeech, with Zipformer-L encoder, CR-CTC achieves the following WER (%) improvements: 2.09/4.59 -> 1.96/4.08 for CTC/AED, 2.00/4.38 -> 1.88/3.95 for pruned transducer.
> - CR-CTC also clearly surpasses the straightforward methods that use an auxiliary head of AED or transducer for jointly training to improve CTC performance (Table 7).
>
> > The paper lack of citation to many very relevant work:
> Most important one: Contrastive siamese network for semi-supervised speech recognition (https://arxiv.org/pdf/2205.14054). The paper focus on compare with SoTA, instead of compare with literature self-distill or other consistency based baseline. In that paper, it include practical trick to make SimSiam type of model work for ASR.
>
> We would like to clarify the **main distinctions between our work and the self/semi-supervised ASR works using consistency regularization:**
> - For the self/semi-supervison works, such as C-Siam and Speech SimCLR, consistency regularization is employed as unsupervised objective to train transformer encoder on unlabeled speech data. These works primarily focus on addressing the training issue of the shortcut learning problem such as via reconstruction loss in Speech SimCLR and temporal augmentation in C-Siam. **In contrast, our work focuses on a fully supervised setting, where we use the consistency loss as a regularization term to improve performance of CTC model trained on labeled data. Since the consistency regularization is enforced on CTC distributions, which are stably supervised by the main CTC loss, it inherently avoids the training issues associated with the unsupervised objectives as in Speech SimCLR and C-Siam.**
> - Moreover, when applying the consistency regularization on CTC distributions, we provide indepth insights into its essential behaviors from different perspectives: **self-distillation, masked prediction which learns contextual representations, and peak suppression which mitigates overfitting and improves the model’s generalization ability.** These are empirically validated by ablation studies in Section 4.3 (Table 4, 5, 6). **Notably, this is the first work to indentify that simply suppressing the peaky CTC distributions can clearly improve the CTC performance,** as demonstrated by our additionally proposed Smooth-Regularized CTC (SR-CTC), which is specifically designed to learn smoother CTC distributions (Appendix Section A.1).

---

> ### Author Response · Authors · 2024-11-21
> **Thanks and response to concerns (part 2)**
>
> > Please properly cite literature work and make proper comparison.
>
> Thank you for pointing out the missing references. We have made the following updates in the first revised version:
>
> - In the revised version of the manuscript, we have **updated the paragraphs discussing related works on consistency regularization in Section 2 to more explicitly clarify the distinctions of our work.** Additionally, while we have already cited many relevant references in the original version, we have now **added the following previously omitted ones:**
>   - Khorram, Soheil, et al. "Contrastive siamese network for semi-supervised speech recognition." ICASSP 2022-2022 IEEE International Conference on Acoustics, Speech and Signal Processing (ICASSP). IEEE, 2022. (mentioned by Reviewer LMc2)
>   - Sapru, Ashtosh. "Using data augmentation and consistency regularization to improve semi-supervised speech recognition." (2022). (mentioned by Area Chair bFCD)
>   - He, Kaiming, et al. "Momentum contrast for unsupervised visual representation learning." Proceedings of the IEEE/CVF conference on computer vision and pattern recognition. 2020.
>   - Jiang, Dongwei, et al. "Speech simclr: Combining contrastive and reconstruction objective for self-supervised speech representation learning." arXiv preprint arXiv:2010.13991 (2020).
>   - Weninger, Felix, et al. "Semi-supervised learning with data augmentation for end-to-end ASR." arXiv preprint arXiv:2007.13876 (2020)
> - In response to the comment, we have **supplemented an additional self-distillation experiment in Section 4.3.** Specifically, we construct a teacher model by tracking the model weights using exponential moving average (EMA), and incorporates an auxiliary loss to learn from the CTC distribution of the teacher model. We refer to this method as **EMA-distilled CTC**, with details provided in Appendix Section A.6. **Experimental results in Table 4 show that CR-CTC significantly outperforms EMA-distilled CTC (2.12/4.62 vs. 2.31/5.25).**

---

> > ### Comment · Reviewer_LMc2 · 2024-11-24
> > **Thanks for the detailed reply!**
> >
> > Really appreciate the author update the related work, I've updated my score.

---

> > > ### Author Response · Authors · 2024-11-25
> > > **Thanks!**
> > >
> > > Thank you very much for your valuable feedback and for improving the score!

---

### Author Response · Authors · 2024-12-04
**Summary of our revisions**

We would like to express our sincere gratitude to all the reviewers for their insightful comments and constructive feedback, which have significantly contributed to improving the quality of our paper. We also greatly appreciate the guidance provided by the Area Chair. Initially, we received scores of 3, 5, 8, 8, which have improved to 6, 5, 8, 8 after the rebuttal process.

Below, we provide a summary of the revisions made in response to the reviewers' suggestions.

* We have updated the paragraphs discussing related work on consistency regularization in Section 2 to more clearly highlight the distinctions between our work and existing approaches. (In response to Reviewer LMc2 and Area Chair bFC)
* We have added an additional self-distillation experiment, EMA-distilled CTC, in Section 4.3. The experimental results in Table 4 demonstrate that CR-CTC significantly outperforms EMA-distilled CTC (2.12/4.62 vs. 2.31/5.25). (In response to Reviewer LMc2)
* Reviewers 6WyR and Reviewer HiGE expressed concerns that the improvements observed on the 10k-hour GigaSpeech dataset were smaller than those on the 1k-hour LibriSpeech dataset. First, we clarified that this is expected, as regularization methods typically show smaller gains when using larger amount of training data due to a reduced tendency for overfitting. Second, we also conducted additional experiments on the 10k-hour WenetSpeech dataset and the 50k-hour LibriHeavy dataset. The results demonstrate that the improvements are still substantial: 7.73/10.83 -> 6.63/8.63 with WenetSpeech and 2.14/4.66 -> 1.92/3.58 with LibriHeavy. We believe these significant improvements further validate the effectiveness and generalization ability of our approach, and we hope they address the reviewers' concerns.
* We have added experiments on the LibriSpeech dataset using a Conformer encoder in Appendix A.7. The results demonstrate that our CR-CTC is also effective with Conformer, significantly outperforming standard CTC and achieving slightly better results compared to CTC/AED and the transducer model. (In response to Reviewer HiGE and Reviewer LPhF)
* In response to the comments, we have corrected typos, adjusted the table positions to optimize space usage, and clarified the omitted implementation details.

---

### Meta-Review · Area_Chair_bFCD · 2024-12-19

**Metareview:**

This paper proposed to use consistency regulation to improve CTC training by enforcing constancy between two CTC distributions obtained from different augmented views of the input speech Mel-spectrogram.  The proposed method is simple but effective, showing impressive improvements from solid baseline on Librispeech, Aishell-1, and Gigaspeech.

Reviewers questioned the novelty, scalability, and applicability of the method to other architectures beyond Zipformer. The authors rebutted by highlighting their focus on supervised training as opposed to self-supervised training in existing literature. They added more discussion of the related works, including missing references in the initial submission. To address architectural concerns, further experiments with Conformer showed similar improvements. Additionally, tests on the 10k-hour WenetSpeech dataset demonstrated convincing gains, alleviating scalability concerns.

Despite some reservations about the novelty, I concur with reviewers that the method is straightforward and effective. Considering the authors addressed reviewers’ concerns during rebuttal, the paper has the value for publication.

In summary, the strength of this paper is that it presents a simple but effective method to improve CTC performance by regularizing the constancy between two  CTC distributions obtained from different augmented views  of the input speech Mel-spectrogram. The weakness of the paper is that the consistency regularization has been studied in the literature (e.g., for self-supervised RNN-T training) although there is no work for supervised CTC training.

**Additional Comments On Reviewer Discussion:**

Reviewers questioned the novelty, scalability, and applicability of the method to other architectures beyond Zipformer. The authors rebutted by highlighting their focus on supervised training as opposed to self-supervised training in existing literature. They added more discussion of the related works, including missing references in the initial submission. As a result, one reviewer raised the score from 3 to 6.

To address architectural concerns, further experiments with Conformer showed similar improvements. Additionally, tests on the 10k-hour WenetSpeech dataset demonstrated convincing gains, alleviating scalability concerns.

Overall, the authors addressed reviewers’ concerns during rebuttal. Therefore, despite some reservations about the novelty, I tend to accept the paper.

---

### Decision · Program_Chairs · 2025-01-22

Accept (Poster)